# Observation of persister cell histories reveals diverse modes of survival in antibiotic persistence

**Miki Umetani[1,2,3]\*[†‡], Miho Fujisawa[3†], Reiko Okura[3], Takashi Nozoe[1,2,3], Shoichi Suenaga[4], Hidenori Nakaoka[5], Edo Kussell[6,7], Yuichi Wakamoto[1,2,3]\***

[1]Research Center for Complex Systems Biology, The University of Tokyo, Tokyo, Japan; [2]Universal Biology Institute, The University of Tokyo, Tokyo, Japan; [3]Department of Basic Science, Graduate School of Arts and Sciences, The University of Tokyo, Tokyo, Japan; [4]Department of Neuropathology, Graduate School of Medicine, The University of Tokyo, Tokyo, Japan; [5]Department of Optical Imaging, Advanced Research Promotion Center, Tokushima University, Tokushima, Japan; [6]Department of Biology, New York University, New York, United States; [7]Department of Physics, New York University, New York, United States

**\*For correspondence:**
miki_umetani@cell.c.u-tokyo.ac.jp (MU);
cwaka@mail.ecc.u-tokyo.ac.jp (YW)

[†]These authors contributed equally to this work

**Present address:** [‡]Department of Biology, New York University, New York, United States

**Competing interest:** The authors declare that no competing interests exist.

**Abstract** Bacterial persistence is a phenomenon in which a small fraction of isogenic bacterial cells survives a lethal dose of antibiotics. Although the refractoriness of persistent cell populations has classically been attributed to growth-inactive cells generated before drug exposure, evidence is accumulating that actively growing cell fractions can also generate persister cells. However, single-cell characterization of persister cell history remains limited due to the extremely low frequencies of persisters. Here, we visualize the responses of over one million individual cells of wildtype *Escherichia coli* to lethal doses of antibiotics, sampling cells from different growth phases and culture media into a microfluidic device. We show that when cells sampled from exponentially growing populations were treated with ampicillin or ciprofloxacin, most persisters were growing before antibiotic treatment. Growing persisters exhibited heterogeneous survival dynamics, including continuous growth and fission with L-form-like morphologies, responsive growth arrest, or post-exposure filamentation. Incubating cells under stationary phase conditions increased both the frequency and the probability of survival of non-growing cells to ampicillin. Under ciprofloxacin, however, all persisters identified were growing before the antibiotic treatment, including samples from post-stationary phase culture. These results reveal diverse persister cell dynamics that depend on antibiotic types and pre-exposure history.

## Editor's evaluation

The manuscript reports on the single cell evaluation of *E. coli* persisters under antibiotics. Typically, persisters are rare cells and only very few have been directly observed in non mutated strains. Therefore, the current work adds an important contribution by mapping a higher numbers of persisters and by convincingly identifying different persister phenotypes. The main conclusions are along previous reports, namely that persisters under β-lactams are mainly non-growing cells from stationary conditions, but if cultures are carefully kept away from stationary phase conditions, the few residual persisters observed in this study display different phenotypes involving growing cells and L-forms.

## Introduction

Antibiotics are potent chemotherapeutic substances that are heavily relied upon to counter bacterial infectious diseases. Nonetheless, bacterial cells often circumvent elimination and continue to survive under antibiotic exposure by acquiring genetic changes to enhance their resistance or tolerance, or by producing less susceptible phenotypic variants. Bacterial persistence is a phenomenon in which a small subset of cells in an isogenic cell population survives for a prolonged period under exposure to lethal doses of antibiotics (*Levin and Rozen, 2006*; *Dhar and McKinney, 2007*; *Lewis, 2007*; *Jayaraman, 2008*; *Balaban, 2011*; *Maisonneuve and Gerdes, 2014*; *Kester and Fortune, 2014*; *Van den Bergh et al., 2017*). The phenomenon occurs across a wide range of combinations of bacterial species and antibiotics, including clinically important pathogens (*Van den Bergh et al., 2017*), such as *Mycobacterium tuberculosis* (*Gomez and McKinney, 2004*; *Dhar and McKinney, 2007*; *Keren et al., 2011*), *Staphylococcus aureus* (*Lechner et al., 2012*; *Johnson and Levin, 2013*), and *Pseudomonas aeruginosa* (*Liebens et al., 2014*), and consequently, it has been recognized as a global health concern (*Gomez and McKinney, 2004*; *Levin and Rozen, 2006*; *Dhar and McKinney, 2007*; *Lewis, 2007*; *Jayaraman, 2008*; *Van den Bergh et al., 2017*).

Since the first elaboration of persistence (*Bigger, 1944*), growth-arrested cells generated before drug exposure (often referred to as 'dormant cells') have been thought to be responsible for the persistence of cellular populations (*Levin and Rozen, 2006*; *Dhar and McKinney, 2007*; *Lewis, 2007*; *Jayaraman, 2008*; *Balaban, 2011*; *Maisonneuve and Gerdes, 2014*; *Kester and Fortune, 2014*; *Van den Bergh et al., 2017*). This explanation of persistence was based on the observations that most antibiotics were ineffective to bacterial cell populations under growth-inhibiting conditions, such as low temperature, low nutrients, and presence of bacteriostatic substances (*Bigger, 1944*). Persistence caused by such prior exposure to stressful environments is now called *triggered persistence* (*Balaban et al., 2019*). However, examining this hypothesis by directly observing individual cell lineages is challenging due to extremely low frequencies of surviving cells (typically $10^{-6}$-$10^{-3}$ [*Keren et al., 2004*]). Consequently, the observation is limited to high persistence mutants (*Balaban et al., 2004*; *Svenningsen et al., 2019*), to naturally high-persisting combinations of bacterial species and drugs (*Wakamoto et al., 2013*; *Goormaghtigh and Van Melderen, 2019*), or to conditions that significantly increase the frequencies of surviving cells (*Bamford et al., 2017*; *Pu et al., 2019*; *Bakshi et al., 2021*). Several studies have suggested that persistence occurs without growth-arrested cells (*Wakamoto et al., 2013*; *Orman and Brynildsen, 2013*; *Goormaghtigh and Van Melderen, 2019*). For example, Wakamoto et al. conducted single-cell observation of persistence of *Mycobacterium smegmatis* against isoniazid (INH), revealing that the surviving cells grew normally before drug exposure and also grew slowly under drug exposure (*Wakamoto et al., 2013*). Using a cell-sorting assay, Orman and Brymildsen showed that even the fast-growing fraction of an exponentially growing *E. coli* population produced cells that survived lethal doses of ampicillin and ofloxacin (*Orman and Brynildsen, 2013*). Furthermore, Goormaghtigh and Van Melderen showed by single-cell observation that *E. coli* persisters against ofloxacin in exponential cultures were dividing normally before exposure (*Goormaghtigh and Van Melderen, 2019*). These results imply that bacterial persistence occurs even without growth-arrested cells generated prior to antibiotic treatment.

To investigate how different dynamics of survival to antibiotic treatment emerge at the single-cell level depending on the past growth history, culture media, and antibiotics, we visualized the response of more than $10^6$ individual cells to antibiotics exposure utilizing a microfluidic device and analyzed the persistence of wildtype *E. coli* (MG1655 strain), one of the best-studied model systems of bacterial persistence. We reveal that most of the surviving cells for which we could identify the single-cell history were growing actively before drug exposure when cell populations sampled from exponential phase were exposed to the $\beta$-lactam ampicillin (Amp) or the fluoroquinolone ciprofloxacin (CPFX). In the case of Amp treatment, these growing persister cells showed heterogeneous responses to treatment, ranging from cells that stopped growing after the first division under drug exposure to those that continued to grow and divide with L-form-like morphological changes. When cells were sampled from stationary phase, allowed to regrow in the microfluidic device with fresh medium flowing for several hours, and treated with an antibiotic drug, most persister cells that gave rise to proliferating progeny cell populations were derived from the non-growing cell fraction in the case of Amp. However, in the case of CPFX treatment, all persister cells for which we identified the single-cell history were growing

before treatment. Therefore, by tracking persister cells before and after antibiotic exposure under different conditions, we reveal multiple survival dynamics underlying antibiotic persistence.

## Results

### The MCMA device reveals single-cell dynamics of persister cells

To visualize the dynamics of low-frequency persister cells, we used a microfluidic device equipped with a membrane-covered microchamber array (MCMA; *Figure 1*). *E. coli* cells were enclosed in the 0.8-µm deep microchambers etched on a glass coverslip by covering microchambers with a cellulose semipermeable membrane via biotin-streptavidin bonding (*Figure 1A–C*; *Inoue et al., 2001*). *E. coli* cells grow in a monolayer and form two-dimensional microcolonies in the microchambers (*Figure 1B–D*). We can control medium conditions around cells flexibly by the medium flow above the membrane (*Figure 1B–D*). The medium exchange rate across the membrane has been evaluated in a similar device previously (*Shimaya et al., 2021*); it is reported that the medium in the microchamber is exchanged within 5 min, which is sufficiently shorter than the periods of antibiotic treatment and the periods of post-antibiotic treatment of observing regrowth in this study. Similar microfluidic devices with membrane-covered microchambers have been utilized for analyzing bacterial growth and starvation (*Inoue et al., 2001*; *Umehara et al., 2003*; *Shimaya et al., 2021*).

We mainly used two *E. coli* strains in the single-cell analysis: MG1655, a strain derived from a wild-type K-12 isolate of *E. coli* (*Blattner et al., 1997*); and MF1, an MG1655-derived strain that expresses RpoS-mCherry from the native *rpoS* locus on the chromosome and green fluorescent protein (GFP) from a low copy plasmid (*Zaslaver et al., 2006*). RpoS is a specialized sigma factor controlling the general stress response (*Battesti et al., 2011*). We originally constructed the MF1 strain to study the correlation between the expression levels of RpoS and the persister trait at the single-cell level. However, a previous report clearly demonstrated that the fluorescent fusion of RpoS is functionally defective (*Patange et al., 2018*). We also confirmed this functional issue of RpoS of MF1; although the dependence on growth phase and culture media of RpoS-mCherry expression was similar to that characterized previously using *lacZ* fusion (*Figure 1—figure supplement 1A and B*; *Lange and Hengge-Aronis, 1994*), this strain was significantly more sensitive to $H_2O_2$ oxidative stress than MG1655 and as sensitive as the MG1655 Δ*rpoS* strain (*Figure 1—figure supplement 1C*). Therefore, we considered MF1 as a strain with impaired RpoS function. Despite this difference, the minimum inhibitory concentrations (MICs) against Amp and CPFX were almost identical between MG1655 and MF1 (*Figure 1—figure supplement 2*).

The population killing curves of the MG1655 and MF1 strains (*Figure 1—figure supplement 3*) exhibit biphasic or multiphasic decay when exponentially growing cell populations are treated with 200 µg/mL of Amp (12.5×MIC, *Figure 1—figure supplement 2A*; see *Figure 1—figure supplement 4* for the growth curves of these strains) or with 1 µg/mL of CPFX (32×MIC, *Figure 1—figure supplement 2B*), confirming that these *E. coli* strains exhibit persistence against these antibiotics (*Balaban et al., 2019*). In the case of Amp treatment, the frequency of surviving cells of MG1655 was higher than that of MF1 up to 3 h of treatment, but became lower in the treatment longer than 3 h (*Figure 1—figure supplement 3A*) due to the delayed shift of the decay rate. Similarly, in the case of CPFX treatment, the frequency of surviving cells of MG1655 was slightly higher than that of MF1 up to 1 h, but then became lower at least up to 7 h of treatment (*Figure 1—figure supplement 3B*). These results confirm that intact RpoS function is not a prerequisite for the occurrence of persistence, but may affect the frequency of surviving cells and the timing of the decay rate shift.

*Figure 2* shows examples of single-cell observation of persister cells using the MCMA device. In this observation, we monitored 110–170 microchambers with a 100× objective in each time-lapse experiment to observe individual cells of the MF1 strain sampled from exponential-phase batch cultures ('post-exponential phase'; *Figure 1C and D*, *Figure 1—figure supplement 4*) with a high spatial resolution. After initial pre-culturing periods of 1.5 h without Amp in the device, we exposed cell populations to 200 µg/mL of Amp for 3.5 h. The Amp exposure lysed most cells (*Figure 2*, *Videos 1 and 2*). However, we detected two persister cells that survived and regrew after the Amp exposure in the ten repeated experiments conducted with the MCMA device. Interestingly, one of the persister cells of MF1 grew and divided actively before the Amp exposure (*Figure 2A-D* and *Video 1*). In response to the Amp exposure, the cell radically changed its cell morphology from a rod shape to a round shape

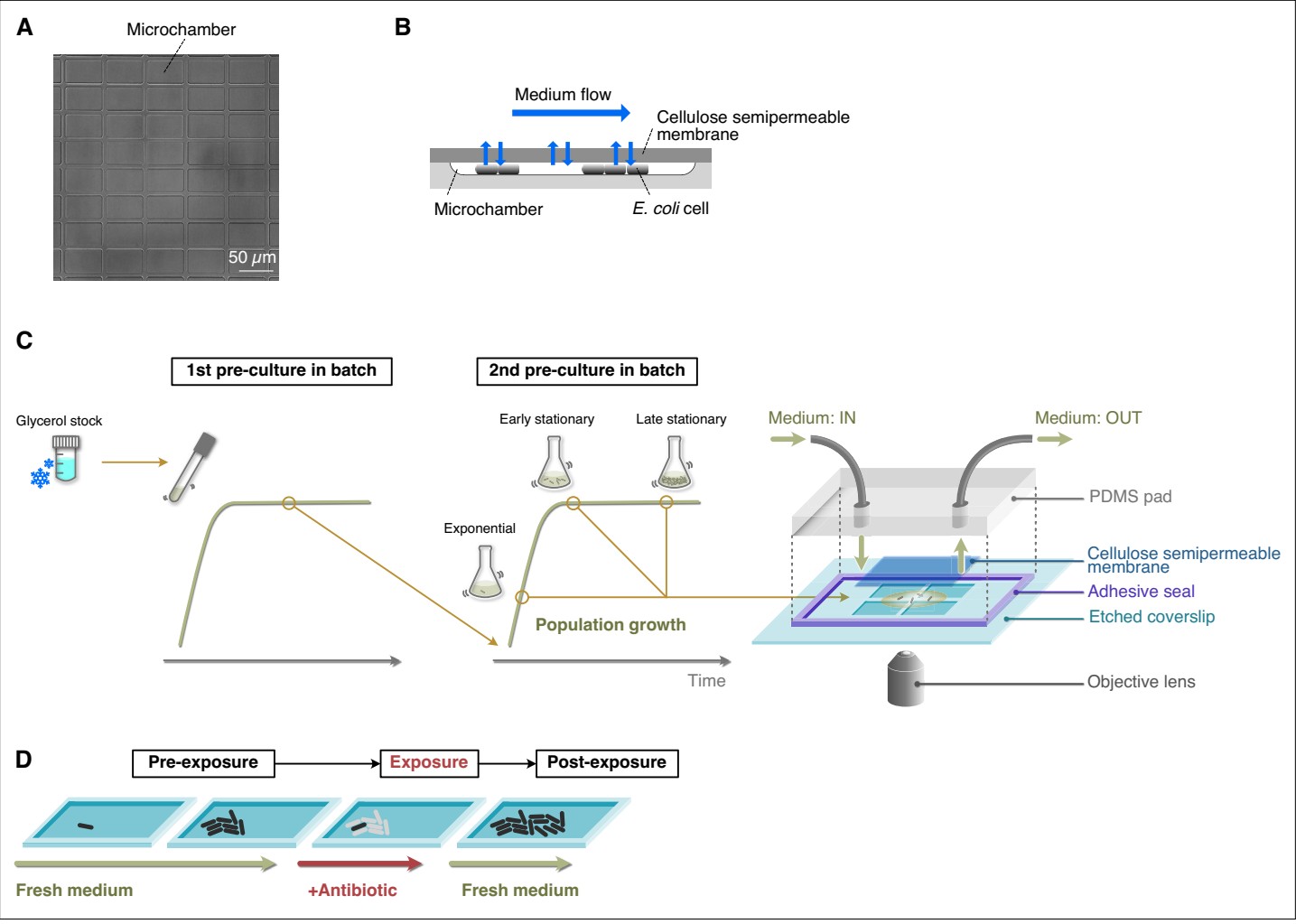

**Figure 1.** Experimental procedures for single-cell time-lapse measurements using the MCMA device. (**A**) A micrograph of the microchamber array etched in a glass coverslip. (**B**) Schematic drawing of cell cultivation in the MCMA device. Cells are enclosed in the microchambers by a cellulose semipermeable membrane cover. A culture medium flows above the membrane. The medium in the microchambers is exchanged through diffusion across the membrane. Cells grow in a monolayer until they are thoroughly packed in the microchambers. (**C**) Loading *E. coli* cells in the MCMA device. *E. coli* cells from a glycerol stock were inoculated into a fresh medium and incubated overnight at 37°C with shaking (first pre-culture). The first pre-culture was diluted to the pre-determined cell density in the fresh medium, and incubated again at 37°C with shaking (second pre-culture). We sampled cells from the second pre-culture at the pre-determined time points for each experiment (exponential phase, early stationary phase, and late stationary phase, see *Figure 1—figure supplement 4*). A suspension of *E. coli* cells sampled from the batch culture was placed on the microchamber array etched in a coverslip. Placing a semipermeable membrane on the cell suspension randomly seeds the cells in the microchambers. The semipermeable membrane tightly seals the microchambers via the biotin-streptavidin binding. A PDMS pad attached to the coverslip via an adhesive frame seal chamber allows medium perfusion. The scale of the microchambers is exaggerated in this diagram for an intuitive explanation. (**D**) Sequence of environmental conditions in time-lapse measurements. *E. coli* cells enclosed in the microchambers were first grown under the flow of fresh medium (pre-exposure period) and exposed to an antibiotic by switching the flowing medium to the antibiotic-containing medium (exposure period). After the pre-determined exposure period, the flowing medium was switched back to the antibiotic-free medium, and the surviving cells were allowed to proliferate again (post-exposure period).

The online version of this article includes the following figure supplement(s) for figure 1:

**Figure supplement 1.** Evaluation of the RpoS expression and the sensitivity to oxidative stress of the MF1 strain.

**Figure supplement 2.** MIC tests.

**Figure supplement 3.** Comparison of the killing curves between MG1655 and MF1.

**Figure supplement 4.** Population growth curves of MG1655 and MF1.

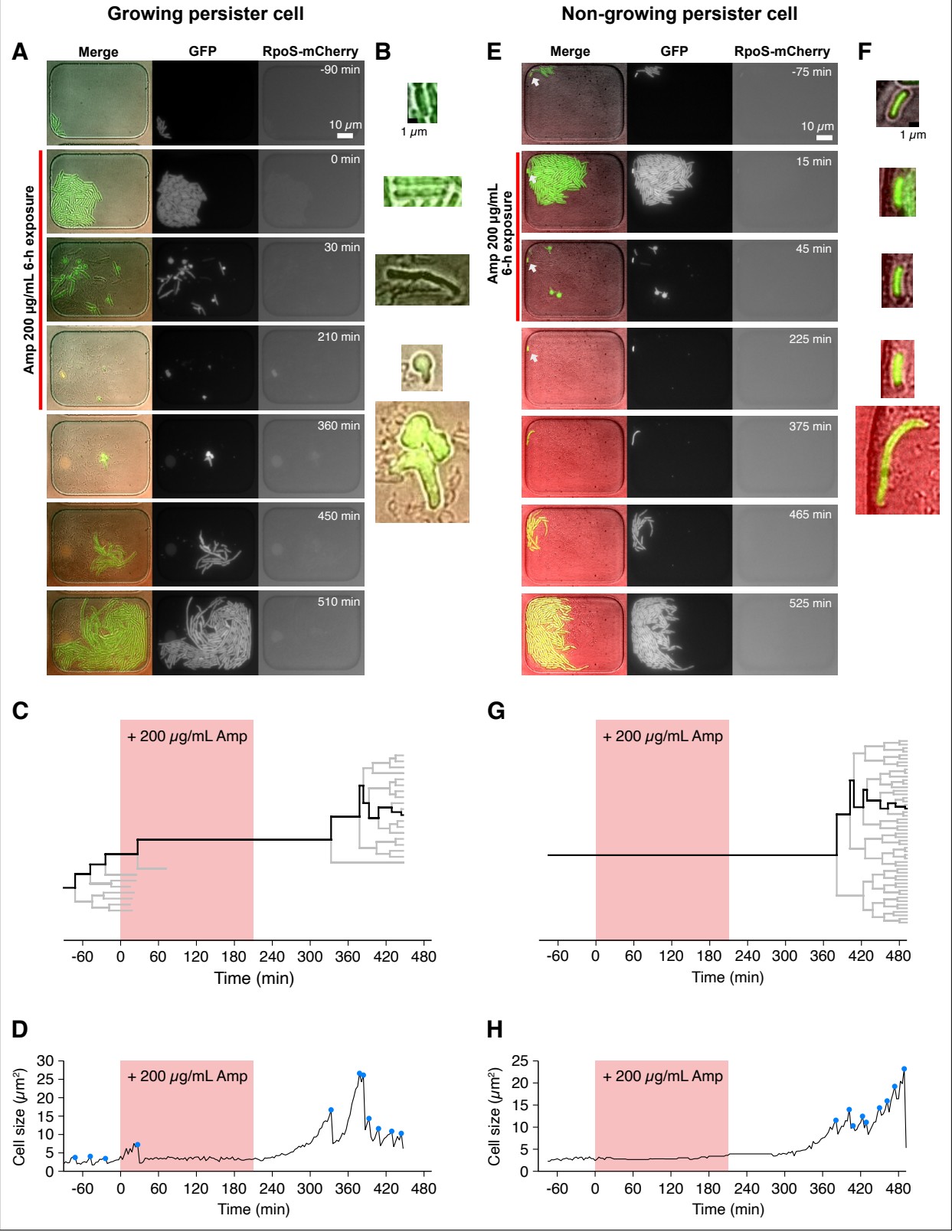

**Figure 2.** Detecting single-cell histories of persister cells using the MCMA device. (**A**) Time-lapse images of a growing persister cell of MF1 against Amp in the post-exponential phase cell populations in LB. The right and middle image sequences show the RpoS-mCherry and GFP fluorescence images. The left image sequence shows the merged images of bright-field (grayscale), GFP (green), and RpoS-mCherry (red) channels. The red line to the left of the images indicates the images under Amp exposure. (**B**) Enlarged micrographs of the growing persister cell in A. (**C**) Single-cell lineage

*Figure 2 continued on next page*

*Figure 2 continued*

tree that produced the growing persister cell. The red background indicates the 3.5 h period of the 200 μg/mL Amp exposure. The bifurcations of the branches represent cell divisions. The endpoints of the branches under the Amp exposure represent cell lysis events. The thick black line indicates the cell whose cell size is shown in D. (**D**) Cell size transition of the growing persister cell. The blue dots indicate the points where cell divisions or cell body fissions occurred. (**E**) Time-lapse images of a non-growing persister cell. The correspondence of the image sequences to the acquisition channels is the same as in B. The white arrows indicate the non-growing cell that eventually proliferated after the Amp exposure. (**F**) Enlarged micrographs of the non-growing persistent cell in E. (**G**) Single-cell lineage tree that produced the non-growing persister. (**H**) Cell size transition of the non-growing persister cell indicated by the thick line in G.

with membrane blebbing and transitted to an L-form-like cell (*Errington et al., 2016*; *Figure 2B* and *Video 1*). The other persister cell of MF1 exhibited no growth and division before and during the Amp exposure and restarted growth after drug removal (*Figure 2E–H* and *Video 2*). This sequence of growth behavior is consistent with the expected survival dynamics for persistence caused by pre-existing non-growing cells (*Balaban et al., 2004*). The expression levels of RpoS-mCherry were low in all the observed cells, including the persister cells characterized above (*Figure 2A and E*, *Videos 1 and 2*). The suppression of RpoS expression in exponential phases in LB media is consistent with the literature (*Battesti et al., 2011*; *Jishage and Ishihama, 1995*). These results demonstrate that the MCMA device can unravel the single-cell history of low frequency persister cells over the course of antibiotic pre-exposure, exposure, and post-exposure periods.

Hereafter, we refer to the surviving cells that showed growth and division before drug exposure as *growing persisters* (*Figure 2A–D*) and those that exhibited no growth and division as *non-growing persisters* (*Figure 2E–H*), focusing on growth traits during the pre-treatment period in the MCMA device without antibiotic.

## Growing persisters are predominant in post-exponential phase cell cultures exposed to Amp

To further scale up single-cell observation, we next used a 40× objective and acquired only bright-field images in the time-lapse measurements (*Video 3*). These changes in the microscopy configuration reduced the spatial resolution of acquired images and did not involve acquisition of fluorescence images. However, most single-cell lineages could still be tracked, and the drug response of more than $10^5$ individual cells was observable with a time-lapse interval of 3 min.

We first observed the MG1655 cells sampled from a batch culture in the exponential phase in LB (post-exponential phase condition; *Figure 1—figure supplement 4A*). The cells were initially allowed to grow in the microchambers for about 1.5 h by flowing a fresh LB medium over the membrane (*Figure 1C and D*). Then the cells were exposed to 200 μg/mL Amp for 6 h by flowing an Amp-containing LB medium. After exposure, we switched the flowing medium back to LB and monitored cell regrowth typically for 12 h. We note that when some cells rapidly resumed proliferation after treatment, their progeny filled chambers by growth, eventually pushing up the semipermeable membrane covering the chambers, growing into neighboring

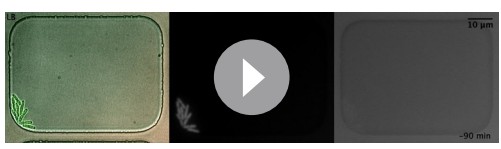

**Video 1.** Time-lapse observation of the growing persister of the MF1 strain against the exposure to 200 μg/mL Amp for 3.5 h. The time-lapse images were acquired with a 100× objective. Left: Merged images of the bright-field (grayscale), GFP (green), and RpoS-mCherry (red) channels; Middle: GFP channel; Right: RpoS-mCherry channel. The medium conditions are indicated near the upper-left corner. The counter near the lower-right corner indicates the time after the onset of Amp exposure. Scale bar, 10 μm.

https://elifesciences.org/articles/79517/figures#video1

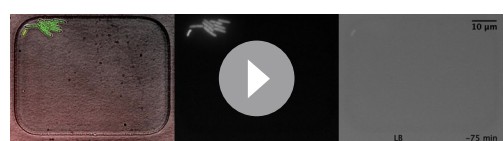

**Video 2.** Time-lapse observation of the non-growing persister of the MF1 strain against the exposure to 200 μg/mL Amp for 3.5 h. The time-lapse images were acquired with a 100× objective. Left: Merged images of the bright-field (grayscale), GFP (green), and RpoS-mCherry (red) channels; Middle: GFP channel; Right: RpoS-mCherry channel. The medium conditions are indicated near the lower-right side. The counter near the lower-right corner (right side of the noted medium conditions) indicates the time after the onset of Amp exposure. Scale bar, 10 μm.

https://elifesciences.org/articles/79517/figures#video2

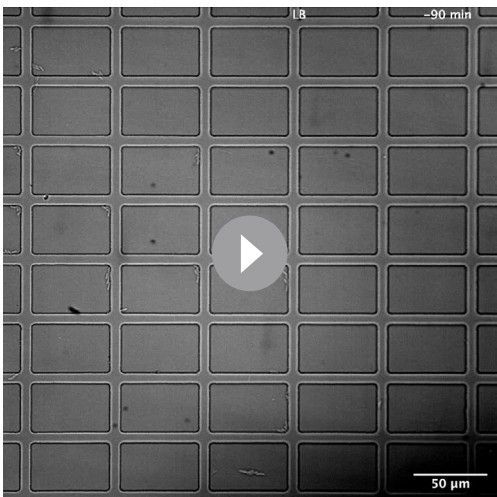

**Video 3.** Simultaneous time-lapse observation of the drug responses of many individual cells using the MCMA microfluidic device. This movie shows a single field of view observed in a time-lapse measurement with the MCMA microfluidic device using a 40× objective. 40 microchambers can be visualized per position. We imaged 50–150 positions in a single time-lapse measurement. In this movie, one microchamber contained a persister cell against Amp exposure (in the upper left microchamber). The medium conditions and the time after the onset of Amp exposure are indicated near the upper-right corner. Scale bar, 50 μm.

https://elifesciences.org/articles/79517/figures#video3

chambers and preventing further observation. In this case, we stopped observing the invaded chambers earlier than 12 h. We set the above duration of antibiotic exposure to be longer than the time of the decay rate shift of the population killing curve (*Figure 1—figure supplement 3A*) to ensure that only persister cells are allowed to re-proliferate after the drug removal.

In this post-exponential phase condition, we detected six persister cells (defined as drug-exposed cells that survived Amp exposure and re-proliferated after drug removal) among $3.8 \times 10^5$ drug-exposed cells (*Figures 3A and 4A–D*, *Figure 4—figure supplement 1* and *Video 4*). All persister cells were actively growing and dividing before the Amp exposure. Since it was reported that persisters of *E. coli hipQ* mutant against Amp grew at suppressed rates in LB medium (0.2~0.3 doublings/h; *Balaban et al., 2004*), we quantified the pre-exposure division rates of the observed cells, finding no significant difference between persister cells and non-surviving cells: 2.36±0.26 doublings/h for growing persisters and 2.26±0.04 doublings/h for non-surviving cells (*Figure 4E*; $p$ = 0.74, Wilcoxon rank sum test). Therefore, these persister cells grew as fast as the non-surviving cells prior to the Amp treatment.

We also analyzed the post-exponential phase cells of MF1 exposed to 200 μg/mL of Amp for 3.5 h after 1.5 h of pre-exposure cultivation in the device and found 12 persister cells among $3.0 \times 10^5$ drug-exposed cells (*Figure 3—figure supplement 1*, *Videos 5 and 6*). The duration of the Amp treatment was again determined to be longer than the time of the decay rate shift of the population killing curve of this strain (*Figure 1—figure supplement 3A*). Analysis of the pre-exposure growth dynamics revealed that 10 of the 12 persister cells were growing persisters and the other two persister cells were non-growing persisters (*Figure 3—figure supplement 1*, *Figure 4—figure supplement 2A*, *Videos 5 and 6*). In contrast to MG1655, the pre-exposure division rate of the growing persisters of MF1 was slightly lower than that of non-surviving cells (2.00±0.22 doublings/h for growing persisters; and 2.48±0.06 doublings/h for non-surviving cells; *Figure 4—figure supplement 2B*; $p$ = 0.011, Wilcoxon rank sum test). However, this division rate of the growing persisters of MF1 was significantly larger than the growth rate of the *hipQ* mutant reported in *Balaban et al., 2004*. Therefore, the growing persisters are dominant among the persisters in the post-exponential phase cell populations of MF1 and grow slightly slower than non-surviving cells before Amp treatment.

The non-growing persisters detected in the post-exponential phase cell populations of MF1 (*Figure 3—figure supplement 1* and *Video 6*) may be long-lagging cells that remained non-growing throughout the period from the previous stationary phase in the first pre-culture to the Amp treatment (*Bakshi et al., 2021*; *Levin-Reisman et al., 2010*), or they may have resumed growth during the second pre-culture but stopped growing spontaneously before the start of time-lapse measurements (see Methods and *Figure 1C* for sample preparation procedures for time-lapse measurements). To test which scenario is more plausible, we extended the duration of the exponential phase by preparing exponential phase cell populations in the second pre-culture from a 1000-fold diluted inoculum of stationary phase cells and allowed them to grow to the same cell density before loading the cells into the MCMA device. With this change in the cell sample preparation procedure, the expected frequency of long-lagging cells at the time of Amp treatment becomes approximately $10^{-8}$, which is well below the detection limit of our single-cell measurements (see Methods). Despite this change, we

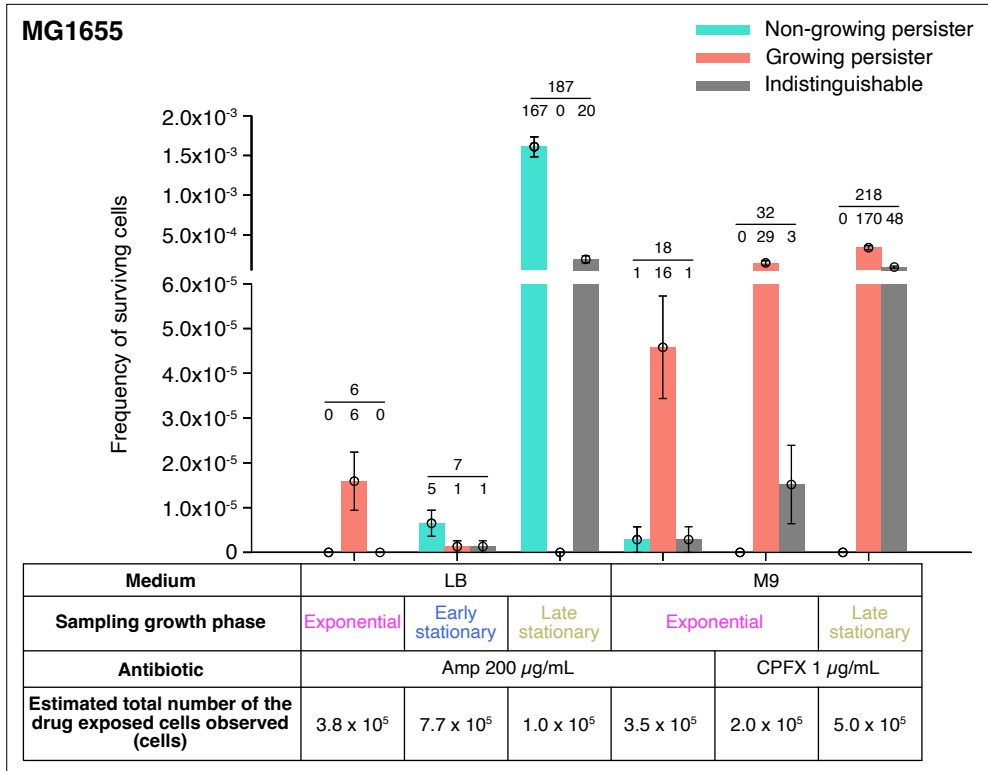

**Figure 3.** Fractions of growing and non-growing persisters of MG1655 against Amp and CPFX with different pre-exposure histories. Green and orange bars represent the frequencies of non-growing and growing persisters in the cell populations detected in the time-lapse measurements of the MG1655 cells. Gray bars show the frequencies of persister cells whose pre-exposure growth traits were indistinguishable. Error bars represent binomial standard errors. The number of cells categorized into each mode is shown above the bar, and the total number of persister cells detected in the time-lapse measurements is shown above the horizontal line. The table shows the estimated total numbers of the drug-exposed cells observed in the time-lapse measurements.

The online version of this article includes the following figure supplement(s) for figure 3:

**Figure supplement 1.** Fractions of growing and non-growing persisters of MF1 against Amp and CPFX with different pre-exposure histories.

still found non-growing persisters at a frequency comparable to the undiluted inoculum experiment: We detected 23 persister cells among $9.9 \times 10^5$ observed cells, and 4 out of 23 persister cells were non-growing persisters (*Figure 3—figure supplement 1A* and *Video 7*). This result shows that non-growing persisters are observed even without the carry-over of lagging cells and are therefore likely to be generated spontaneously in the exponentially growing cell populations (*Balaban et al., 2019*).

## Response of growing persisters to Amp exposure is heterogeneous

To characterize the dynamics of individual persister cells, we quantified transitions in cell size and cell shape (cell body circularity, $4\pi \times [\text{Cell area}]/[\text{Perimeter}]^2$) during pre-exposure, exposure, and post-exposure periods (*Figures 1D and 4A–D*, *Figure 4—figure supplement 1*, and *Figure 4—figure supplement 2A*). The analysis reveals heterogeneous responses of growing persisters of MG1655 to Amp exposure. For example, the growing persister shown in *Figure 4A and B* suppressed growth and stopped division after the first division under the exposure to Amp. This cell resumed division with a long lag approximately 6 h after drug removal (*Figure 4B*). On the other hand, the growing persister shown in *Figure 4C and D* continued to grow throughout the period of Amp treatment with deformation of the cell shape from rod to L-form-like shape. Cell body fission also continued during Amp exposure and gradually restored normal rod shape through multiple fissions after drug removal (*Figure 4C and D*). The other persister cells exhibited intermediate behavior between these two, with growth and cell body deformation in some time periods and growth suppression in other time

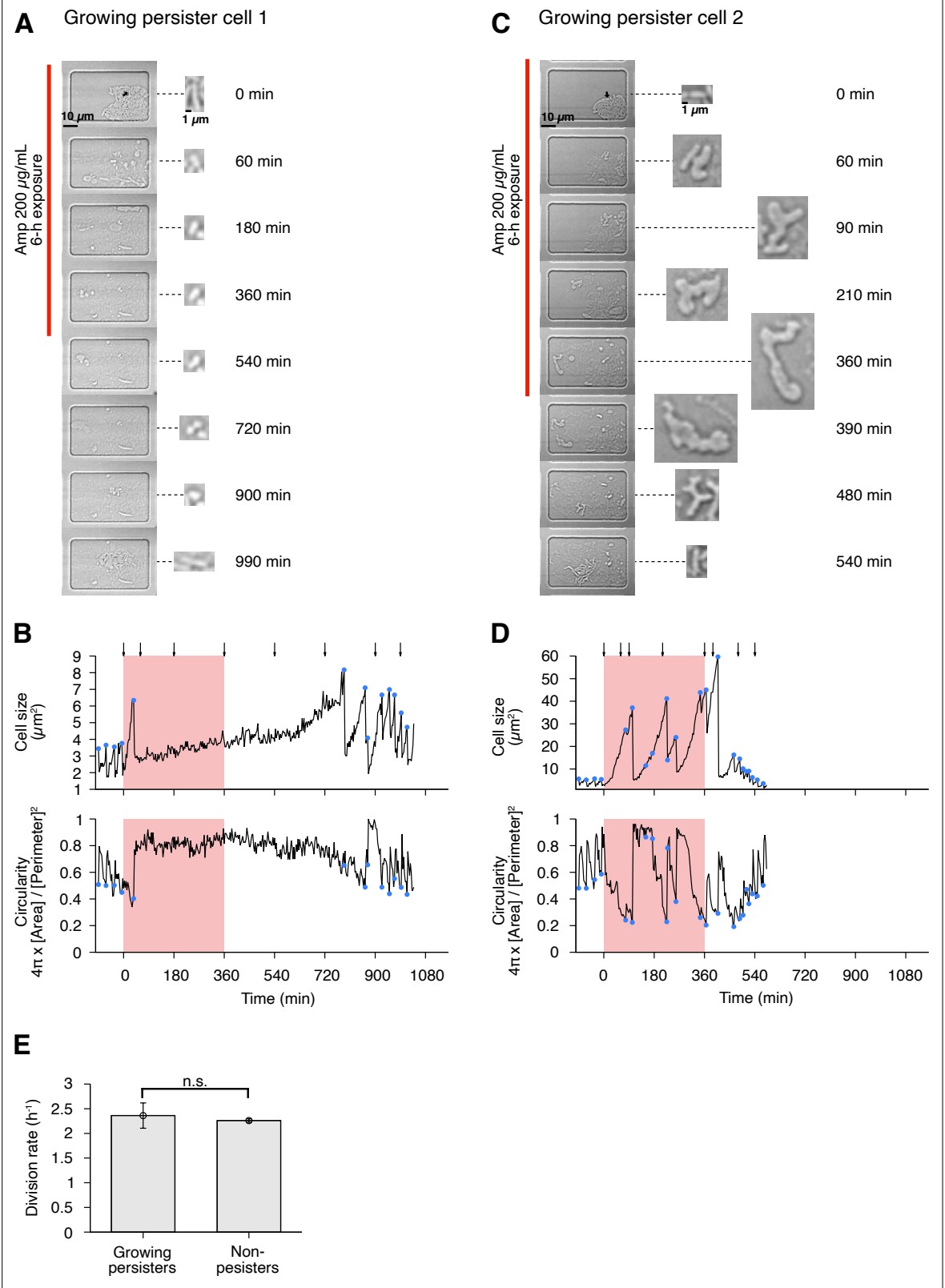

**Figure 4.** Persisters against Amp observed in the post-exponential phase cell populations of MG1655 in LB. (**A**) Time-lapse images of a growing persister whose growth was suppressed during the exposure to 200 µg/mL Amp in LB medium. The left image sequences show the bright-field images. The right image sequences show the enlarged micrographs of the persister cell at each time point. The red line to the left of the images indicates the images under the Amp exposure. The black arrow indicates the growing persister cell. (**B**) Temporal changes in cell size (top graph) and cell shape

*Figure 4 continued on next page*

*Figure 4 continued*

circularity (bottom graph) of the growing persister cell shown in A. The red background indicates the 6-h period of the Amp exposure. The black arrows indicate the time points at which the enlarged micrographs in A were taken. The blue dots indicate the points where cell divisions or cell body fissions occurred. (C) Time-lapse images of a growing persister that continued to grow and divide during the Amp exposure. Images are displayed in the same way as in A. (D) Temporal changes in cell size (top graph) and cell shape circularity (bottom graph) of the growing persister shown in C. (E) Comparison of pre-exposure division rates between growing persisters ($n = 6$) and non-persisters ($n = 204$). Error bars represent standard errors. No significant difference in division rates was detected at the significance level of 0.05 ($p = 0.74$, Wilcoxon rank sum test).

The online version of this article includes the following figure supplement(s) for figure 4:

**Figure supplement 1.** Single-cell dynamics of the MG1655 persisters against Amp in the post-exponential phase cell populations in LB.

**Figure supplement 2.** Single-cell dynamics of the MF1 persisters against Amp in the post-exponential phase cell populations in LB.

periods (*Figure 4—figure supplement 1*). Surprisingly, some of the L-form-like cells crawled around like amoeba cells within the microchambers, detaching and leaving some parts of their cell bodies behind (*Video 4*). Furthermore, rod-shaped regrowing cell populations emerged even from the small and left-behind cell bodies (*Video 4*). These observations reveal the abilities of *E. coli* cells to sustain viability with abnormal morphologies and to restore rod shapes in the absence of agents that induced morphological changes.

The growth and cell shape dynamics of growing persister cells of MF1 were similar to those of MG1655. We detected persister cells that stopped growing and dividing after the first divisions in response to Amp exposure, as well as those that continued growth and cell body fission with deformed cell shape throughout the period of Amp exposure, and those that exhibit intermediate behavior (*Figure 4—figure supplement 2A*). These results reveal heterogeneous responses of growing persisters to Amp treatment with respect to continued growth and division and the occurrence of extensive cell shape changes.

## Growing persisters are also dominant in minimal medium

To test whether medium conditions influence persistence dynamics, as previously suggested (*Harms et al., 2017*), we repeated the experiments with M9 minimal medium. The killing curves of both MG1655 and MF1 cell populations sampled from exponential phase cultures in M9 minimal medium (post-exponential phase condition in M9) show biphasic curves when exposed to the same concentration of Amp as used in LB medium (200 µg/mL), confirming the occurrence of persistence in this condition (*Figure 1—figure supplement 3C–E*; see *Figure 1—figure supplement 4B* for the growth curve). The frequency of surviving cells of MG1655 in M9 was consistently higher than that of MF1 (*Figure 1—figure supplement 3C*). The frequency of persister cells in the post-exponential phase cell populations in M9 was higher than the frequency in the LB medium, approximately 100-fold and 10-fold for MG1655 and MF1, respectively (*Figure 1—figure supplement 3D and E*).

Single-cell time-lapse measurements detected 18 persister cells of MG1655 that regrew and divided after the exposure to 200 µg/mL of

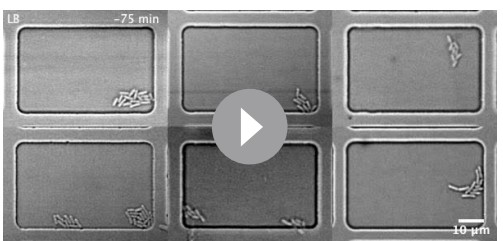

**Video 4.** Time-lapse movie of all six growing persister cells in the post-exponential phase cell populations of MG1655 against the exposure to 200 µg/mL Amp for 6 h in LB. This movie shows all six growing persister cells observed in all the replicated single-cell measurements of the post-exponential phase cells of MG1655 exposed to 200 µg/mL Amp for 6 h in LB (*Figure 3*). We used a 40× objective in this measurement. Scale bar, 10 µm.

https://elifesciences.org/articles/79517/figures#video4

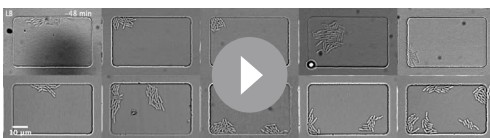

**Video 5.** Time-lapse movie of the 10 growing persisters in the post-exponential phase cell populations of MF1 against the exposure to 200 µg/mL Amp for 3.5 h in LB. This movie shows the 10 growing persister cells observed in all the replicated single-cell measurements of the post-exponential phase cells of MF1 exposed to 200 µg/mL Amp for 3.5 h in LB (*Figure 3—figure supplement 1*). We used a 40× objective in this measurement. Scale bar, 10 µm.

https://elifesciences.org/articles/79517/figures#video5

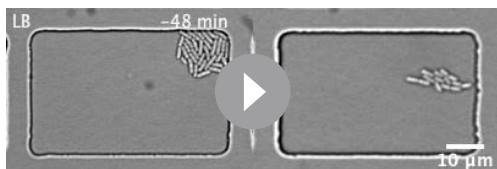

**Video 6.** Time-lapse movie of the two non-growing persisters in the post-exponential phase cell populations of MF1 against the exposure to 200 μg/ mL Amp for 3.5 h in LB. This movie shows the two non-growing persister cells observed in all the replicated single-cell measurements of the post-exponential phase cells of MF1 exposed to 200 μg/mL Amp for 3.5 h in LB (*Figure 3—figure supplement 1*). We used a 40× objective in this measurement. Scale bar, 10 μm. https://elifesciences.org/articles/79517/figures#video6

Amp for 6 h among $3.5×10^5$ drug-exposed cells. Among them, we were able to determine the presence or absence of pre-exposure growth and division of 17 persister cells, of which 16 grew and divided before the Amp exposure, and the other one showed no growth and division (*Figure 3A*, *Figure 5A-D*, *Figure 5—figure supplement 1*, and *Video 8*). The presence or absence of growth and division of 1 persister cell was indeterminate due to translocation of cells within a microchamber. Similarly, we detected 24 persister cells of MF1 that survived and re-proliferated after the exposure to 200 μg/mL of Amp for 6 h among $1.1×10^6$ drug-exposed cells (*Figure 3—figure supplement 1*). The presence and absence of pre-exposure growth and division were identified for 19 persister cells, and we found that 17 of them were growing persisters (*Figure 3—figure supplement 1*, and *Video 9*). No growth or division was observed for the other two persister cells. These results in the M9 minimal medium are qualitatively consistent with those in LB; growing persisters are dominant in the post-exponential phase cell populations for both MG1655 and MF1, and non-growing persisters exist as small fractions.

Despite the consistency of the dominant survival mode of persister cells, the pre-exposure division rate of growing persisters (0.46±0.07 doublings/h) of MG1655 in M9 was significantly lower than that of non-surviving cells (0.96±0.03 doublings/h, $p < 10^{-6}$, Wilcoxon rank sum test; *Figure 5E*). Therefore, growing persisters in the M9 medium tend to grow more slowly than non-surviving cells.

The heterogeneous response of growing persisters to Amp exposure was also observed in the M9 medium; some persister cells suppressed growth and division to varying degrees in response to drug exposure, while others continued to grow and undergo cell body division with a change in cell shape from rod to L-form-like (*Figure 5A–D*, *Figure 5—figure supplement 1*, and *Video 8*). The transition to an L-form-like cell shape was prominent during the return to normal growth and rod shape in the post-exposure period. In contrast to the growing persisters in LB, we observed filamentation of some persister cells in M9 after exposure; normal-sized progeny cells produced from these filamentous cells by unequal cell division regained normal growth (*Figure 5—figure supplement 1* and *Video 8*). Radical morphological changes of cell bodies were observed in many growing cells in the M9 medium, including non-surviving cells (*Video 8*). As observed in the post-exponential phase conditions in LB, some of the L-form-like cells crawled around like amoeba cells within the microchambers, detaching and leaving some parts of their cell bodies behind (*Figure 5C*, *Videos 8 and 10*). Rod-shaped and normally growing progeny cells emerged from these detached cell bodies.

In the single-cell analysis of the persister cells of MG1655 in the M9 medium, we found a single case where a pair of siblings produced during the pre-exposure period both survived the Amp treatment and re-proliferated after the drug removal (growing persister cell 1 in *Figure 5A and B* and growing persister cell 4 in *Figure 5—figure supplement 1*). If the survival of drug-exposed individual cells was determined independently, the probability of finding such a pair of siblings that both survived would be extremely low, and in our measurement virtually zero. Therefore, this

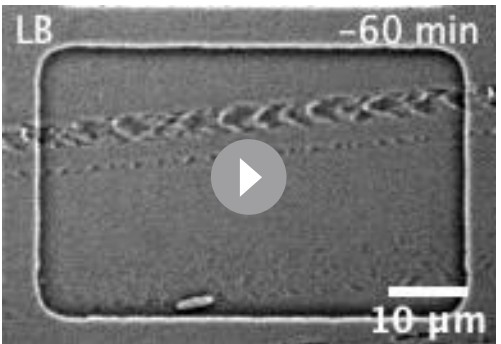

**Video 7.** Non-growing persister of MF1 in the post-exponential phase cell populations grown from a 1000-fold smaller inoculum. This movie shows a representative non-growing persister cell of MF1 found in the post-exponential phase cell population in LB grown from a 1000-fold smaller inoculum to eliminate long lagging cells. We used a 40× objective in this measurement. Scale bar, 10 μm. https://elifesciences.org/articles/79517/figures#video7

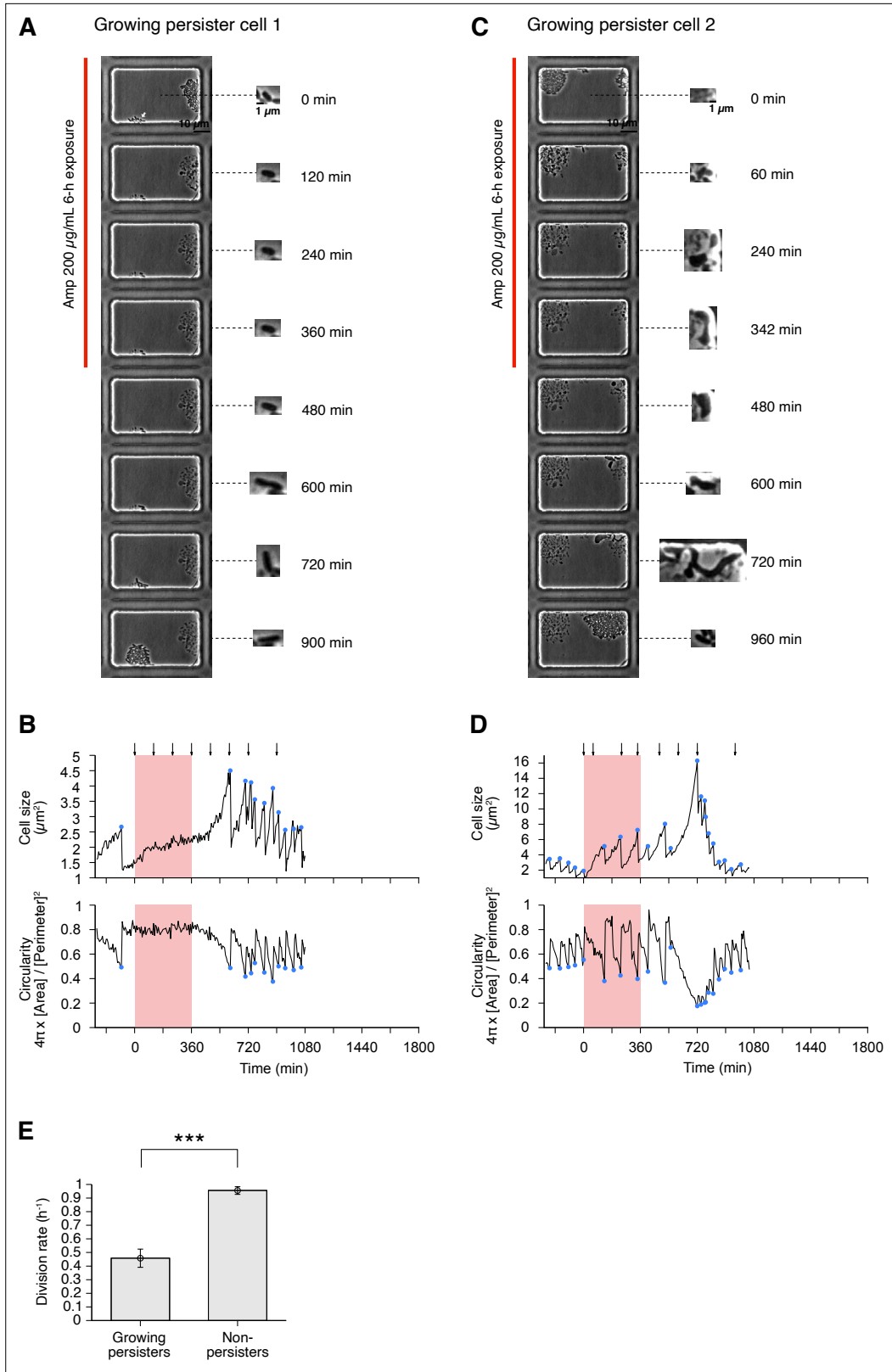

**Figure 5.** Growing persisters against Amp observed in the post-exponential phase cell populations of MG1655 in M9. (**A**) Time-lapse images of a growing persister whose growth was suppressed during the exposure to 200 μg/mL Amp in M9 medium. The left image sequences show the bright-field images. The right image sequences show the enlarged micrographs of the persister cell at each time point. The red line to the left of the images indicates

*Figure 5 continued on next page*

*Figure 5 continued*

the images under the Amp exposure. The white arrow indicates the growing persister cell. (**B**) Temporal changes in cell size (top graph) and cell shape circularity (bottom graph) of the growing persister cell shown in A. The red background indicates the 6-h period of the Amp exposure. The black arrows indicate the time points at which the enlarged micrographs in A were taken. The blue dots indicate the points where cell divisions or cell body fissions occurred. (**C**) Time-lapse images of a growing persister that continued to grow and divide during the Amp exposure. Images are displayed in the same way as in A. (**D**) Temporal changes in cell size (top graph) and cell shape circularity (bottom graph) of the growing persister shown in C. (**E**) Comparison of pre-exposure division rates between growing persisters ($n = 14$) and non-persisters ($n = 106$). Error bars represent standard errors. Due to the difficulty in counting the number of pre-exposure cell divisions, two growing persister cells were excluded from this analysis. A significant difference in division rates was detected between the two groups at the significance level of 0.05 ($p = 9.65 \times 10^{-7}$, Wilcoxon rank sum test).

The online version of this article includes the following figure supplement(s) for figure 5:

**Figure supplement 1.** Single-cell dynamics of the growing persisters of MG1655 against Amp in the post-exponential phase cell populations in M9.

result implies a correlation between cell lineage and the persistence trait, as well as some heritable factors contributing to the survival of individual cells, as previously suggested by a fluctuation test-based batch culture analysis (*Hossain et al., 2023*). However, our observation is limited to this single pair; higher throughput analysis is needed to clarify the cell lineage correlation of survival.

## Stationary phase environments increase the frequencies of non-growing persisters

Several batch-culture assays have shown that stationary phase environments increase the frequencies of persister cells in bacterial cell populations (*Spoering and Lewis, 2001*; *Keren et al., 2004*; *Balaban et al., 2019*). To understand how the past cultivation history affects the survival dynamics of individual persister cells, we sampled cells from early and late stationary phases in LB medium (*Figure 1—figure supplement 4A*), allowed them to grow in the fresh medium for 2 h, and exposed them to 200 µg/mL of Amp in both batch and single-cell cultures (*Figure 1C and D*, *Figure 6A*).

The population killing curves of MG1655 under these conditions (hereafter referred to as 'post-early stationary phase' and 'post-late stationary phase' conditions; *Figure 1—figure supplement 4A*) follow biphasic or multiphasic kinetics under the exposure to 200 µg/mL of Amp (*Figure 6B*). While the killing curve of the post-early stationary phase condition was similar to that of the post-exponential phase condition, that of the post-late stationary phase condition differed significantly from those of the other two conidtions with

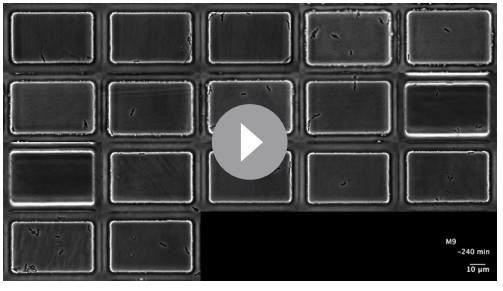

**Video 8.** Time-lapse movie of all 18 persister cells in the post-exponential phase cell populations of MG1655 against the exposure to 200 µg/mL Amp for 6 h in M9. This movie shows all the 18 persister cells observed in all the replicated single-cell measurements of the post-exponential phase cells of MG1655 exposed to 200 µg/mL Amp for 6 h in M9 (*Figure 3*). We used a 100× objective in this measurement. Scale bar, 10 µm.
https://elifesciences.org/articles/79517/figures#video8

**Video 9.** Time-lapse movie of all 24 persister cells in the post-exponential phase cell populations of MF1 against the exposure to 200 µg/mL Amp for 6 h in M9. This movie shows all the 24 persister cells observed in all the replicated single-cell measurements of the post-exponential phase cells of MF1 exposed to 200 µg/mL Amp for 6 h in M9 (*Figure 3—figure supplement 1*). We used a 40× objective in this measurement. Scale bar, 10 µm.
https://elifesciences.org/articles/79517/figures#video9

**Video 10.** Amoeba-like locomotion of L-form-like cells of MF1 in the microchambers. This movie shows representative three microchambers in which L-form-like *E. coli* cells exhibited amoeba-like locomotion. In this experiment, 200 µg/mL of Amp was exposed to the MF1 cells cultured in the M9 medium. Fission of cell bodies occurred in these cells during the locomotion. Furthermore, regrowing cell populations emerged from left-behind cell bodies.

https://elifesciences.org/articles/79517/figures#video10

consistently higher frequencies of surviving cells (*Figure 6B*). The frequencies of surviving cells evaluated at the 6 h time point were $3.7 \times 10^{-7}$ ($1.7 \times 10^{-7}$–$8.1 \times 10^{-7}$) for the post-early stationary phase cell populations and $5.1 \times 10^{-2}$ ($3.9 \times 10^{-2}$–$6.6 \times 10^{-2}$) for the post-late stationary phase cell populations (*Figure 6B*).

Comparison of the killing curves between MG1655 and MF1 shows that for the post-early stationary phase cell populations, the frequencies of surviving cells of MG1655 were higher than those of MF1 up to 2.5 h of Amp exposure, but became lower thereafter, resulting in fewer surviving cells at longer exposures (*Figure 6—figure supplement 1A*). This time-dependent reversal of surviving cell frequencies between MG1655 and MF1 was also observed for the post-exponential phase cell populations, as noted above (*Figure 1—figure supplement 3A*). Under post-late stationary phase conditions, the frequency of surviving cells was consistently higher in MG1655 than in MF1 (*Figure 6—figure supplement 1B*).

To characterize the dynamics of individual persister cells in the cell populations with these different pre-exposure cultivation histories, we next performed single-cell measurements with the MCMA device. We loaded cells in the corresponding stationary phase conditions (*Figure 1—figure supplement 4A*) into the MCMA device and allowed the cells to regrow by changing the media around the cells from the corresponding stationary phase conditioned medium to fresh LB medium at the beginning of the time-lapse measurement (*Figure 1C and D*). After a 2-h pre-exposure period, we exposed the cells to 200 µg/mL of Amp for 6 h and monitored the re-proliferation of surviving cells by flowing drug-free fresh medium (*Figure 1C and D*). Since all cells were maintained in a non-growing state by the corresponding stationary phase conditioned medium until the start of the time-lapse measurement, non-growing cells found in these experiments are by definition lagging cells that failed to rapidly resume growth.

Measurements of MG1655 with the 40× objective detected seven persister cells that survived and re-proliferated after Amp exposure among $7.7 \times 10^5$ drug-exposed cells in the post-early stationary phase cell populations (*Figure 3A* and *Video 11*). We were able to determine the pre-exposure growth states of six persister cells. Among them, one persister cell was a growing persister and the other five were non-growing persisters (*Figure 3A*). Therefore, the proportions of non-growing persisters increased significantly in the post-early stationary phase cell populations despite the similarity of their killing curve to that of post-exponential phase cell populations.

For the post-late stationary phase cell populations, we detected 187 persister cells among $1.0 \times 10^5$ drug-exposed cells, finding that all but 20 indeterminate cells were non-growing persisters (*Figure 3A* and *Video 12*). These results show that entry into stationary phase conditions rapidly changes the predominant pre-exposure trait of persister cells from growing to non-growing. We confirmed that the same conclusion holds true for MF1 persister cells in the post-early and post-late stationary phase cell populations (*Figure 3—figure supplement 1*, *Videos 13 and 14*).

We found that many cells sampled from late stationary phase remained non-growing in fresh medium during the pre-exposure period. Many, but not all, of these long lagging cells also appeared to be intact under Amp exposure, but did not regrow even after the drug removal for at least 12 h. Most of these non-regrowing cells may be viable, as they were negative in propidium iodide (PI) staining (*Figure 6—figure supplement 2* and *Video 15*), although PI is known to be an imperfect marker of cell viability and the possibility that they were actually dead cannot be excluded (*Kim et al., 2018*; *Wu et al., 2024*). These cells may eventually regrow as persisters with long lags after drug removal or may not be able to regrow due to deepened dormancy (*Pu et al., 2019*; *Bollen et al., 2021*; *Bollen et al., 2025*). In fact, it has been shown by the single-cell measurements with a mother machine microfluidic device that the time to regrowth after drug removal is heterogeneous among non-growing cells (*Bakshi et al., 2021*). These late regrowers are detectable in the killing curve assay

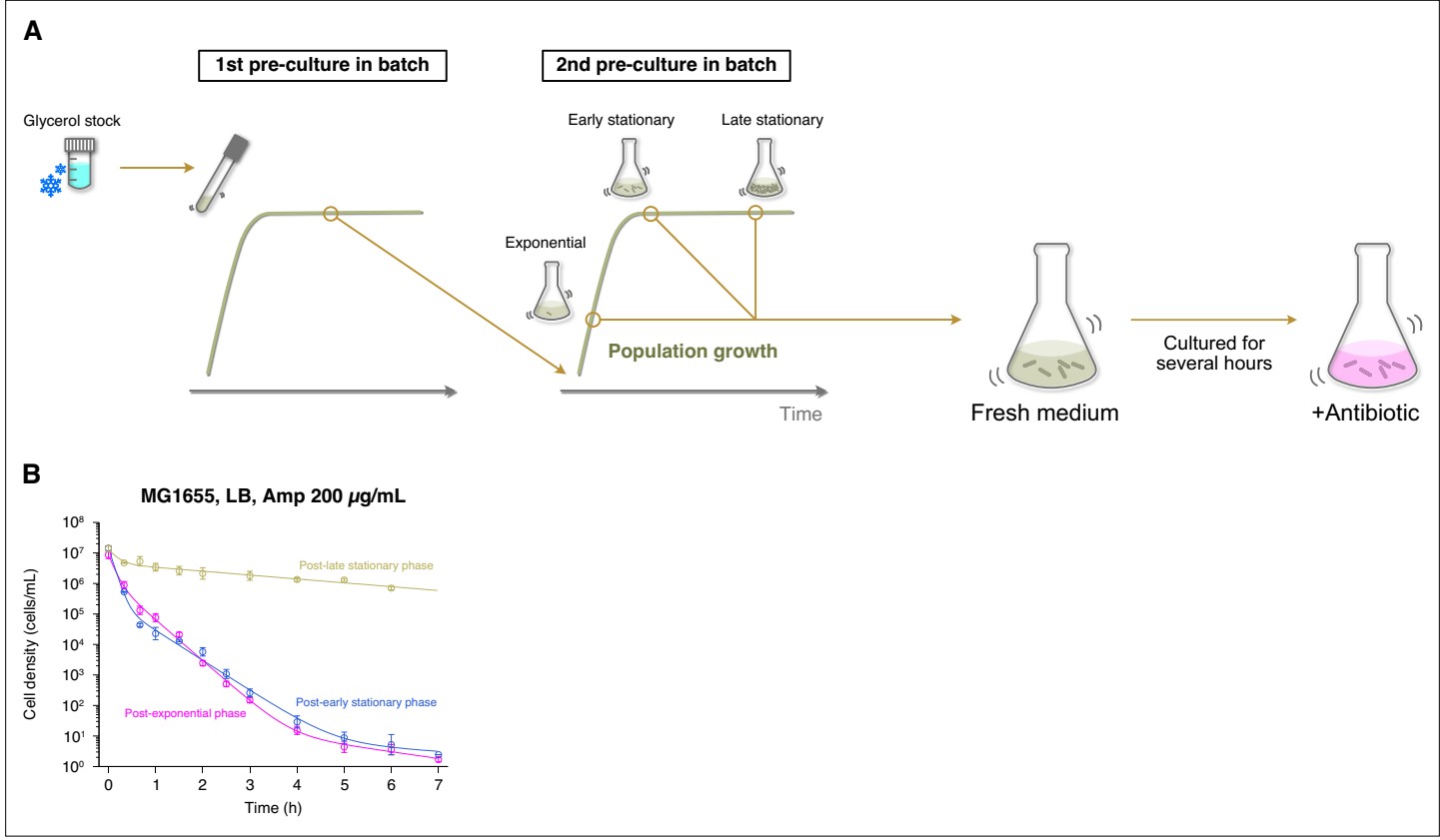

**Figure 6.** Growth phase-dependent persistence dynamics of MG1655 cell populations exposed to Amp in LB medium. (**A**) Experimental procedure for measuring killing curves of post-exponential, post-early stationary, and post-late stationary phase cell populations. *E. coli* cells were inoculated from a glycerol stock into fresh medium and incubated overnight at 37°C with shaking (first pre-culture). The first pre-culture was diluted to the pre-determined cell density in fresh medium and incubated again at 37°C with shaking (second pre-culture). We sampled cells from the second pre-culture at the pre-determined time point for each condition (exponential phase, early stationary phase, or late stationary phase; see *Figure 1—figure supplement 4*). A suspension of *E. coli* cells from the second pre-culture was diluted again in fresh medium, cultured for 2 h, and then treated with 200 µg/mL Amp. We introduced this pre-exposure cultivation in fresh medium to align the pre-exposure history conditions between batch and single-cell measurements (see *Figure 1C*). We collected cell suspensions at the pre-determined time points and estimated the viable cell density in the population by a limiting dilution method (see Methods). (**B**) Killing curves of MG1655 cell populations sampled from different growth phases. The dynamics of post-exponential, post-early stationary, and post-late stationary phase cell populations are shown in magenta, blue, and yellow, respectively. Points and error bars represent means and standard errors of the logarithm of cell density among replicate experiments ($n = 6$ for the post-exponential phase, $n = 3$ for the post-early and post-late stationary phase conditions). The lines represent the fitting of the double or triple exponential decay curves to the data: $N(t) = 7.46 \times 10^6 \exp(-9.89t) + 1.42 \times 10^6 \exp(-3.09t) + 64.1 \exp(-0.508t)$ for the post-exponential phase; $N(t) = 1.46 \times 10^7 \exp(-11.4t) + 2.81 \times 10^5 \exp(-2.27t) + 17.6 \exp(-0.248t)$ for the post-early stationary phase; and $N(t) = 9.07 \times 10^6 \exp(-6.14t) + 4.52 \times 10^6 \exp(-0.292t)$ for the post-late stationary phase, where $N(t)$ is the viable cell density in the cell populations at time $t$. We chose a biphasic or triphasic exponential decay curve based on the Akaike's Information Criterion (AIC) of the fit to the experimental data (see Methods).

The online version of this article includes the following figure supplement(s) for figure 6:

**Figure supplement 1.** Killing curves of post-early stationary and post-late stationary phase cell populations of MF1.

**Figure supplement 2.** Viability check of non-growing cells by PI staining.

but not in the single-cell measurements with the MCMA microfluidic device due to the overgrowth of the early regrowers. Consequently, our single-cell measurements may underestimate the frequency of persister cells in the population sampled from late stationary phase. However, this potentially overlooked fraction of persister cells are also non-growing persisters; the predominance of non-growing persisters in the post-late stationary phase cell population is unchanged.

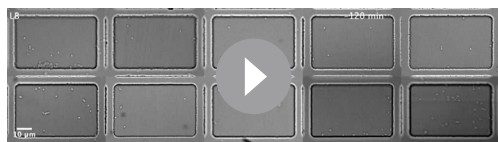

**Video 11.** Time-lapse movie of all 7 persister cells in the post-early stationary phase cell populations of MG1655 against the exposure to 200 µg/mL Amp for 6 h in LB. This movie shows all the seven persister cells observed in all the replicated single-cell measurements of the post-early stationary phase cells of MG1655 exposed to 200 µg/mL Amp for 6 h in LB (*Figure 3*). We used a 40× objective in this measurement. Scale bar, 10 µm.
https://elifesciences.org/articles/79517/figures#video11

**Video 12.** Time-lapse movie of 10 representative persister cells in the post-late stationary phase cell populations of MG1655 against the exposure to 200 µg/mL Amp for 6 h in LB. This movie shows 10 representative persister cells observed in one of the single-cell measurements of the post-late stationary phase cells of MG1655 exposed to 200 µg/mL Amp for 6 h in LB (*Figure 3*). We used a 40× objective in this measurement. Scale bar, 10 µm.
https://elifesciences.org/articles/79517/figures#video12

## Stationary phase environments increase the persister fraction against Amp among non-growing cells

Although the presence of cells that neither grow nor divide during the pre-exposure period in the MCMA device is evident in the post-late stationary phase cell populations, such non-growing cells were also present in the post-exponential and post-early stationary phase cell populations, albeit at lower frequencies. Based on this observation, we next asked how the frequencies of non-growing cells immediately before Amp exposure depend on cultivation history and whether these non-growing fractions produce persisters (non-growing persisters) at comparable frequencies across conditions (*Figure 7A*).

To estimate the fractions of non-growing cells in the populations during the pre-exposure period in the MCMA device, we performed single-cell time-lapse measurements with post-exponential, post-early stationary, and post-late stationary phase cells using the 100× objective. Here, non-growing cells were defined as cells that neither grew nor divided during the 1.5 h period immediately before the Amp exposure in the MCMA device. The estimated fractions of non-growing cells in the MG1655 cell populations immediately before the drug exposure were $(2.9\pm1.5)\times10^{-4}$ ($n = 13650$), $(2.0\pm1.1)\times10^{-3}$ ($n = 1511$), and $0.61\pm0.02$ ($n = 521$) for the post-exponential, post-early stationary, and post-late stationary phase conditions, respectively (horizontal axis in *Figure 7B*). Those in the MF1 cell populations were $(1.6\pm0.9)\times10^{-4}$ ($n = 19122$), $(8.7\pm2.9)\times10^{-3}$ ($n = 1031$), and $0.41\pm0.02$ ($n = 469$) for the post-exponential, post-early stationary, and post-late stationary phase conditions, respectively (horizontal axis in *Figure 7C*). Therefore, the frequency

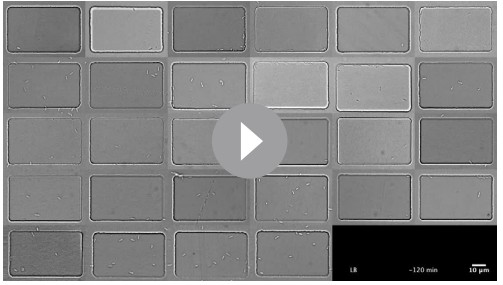

**Video 13.** Time-lapse movie of all 28 persister cells in the post-early stationary phase cell populations of MF1 against the exposure to 200 µg/mL Amp for 3.5 h in LB. This movie shows all the 28 persister cells observed in all the replicated single-cell measurements of the post-early stationary phase cells of MF1 exposed to 200 µg/mL Amp for 3.5 h in LB (*Figure 3—figure supplement 1*). We used a 40× objective in this measurement. Scale bar, 10 µm.
https://elifesciences.org/articles/79517/figures#video13

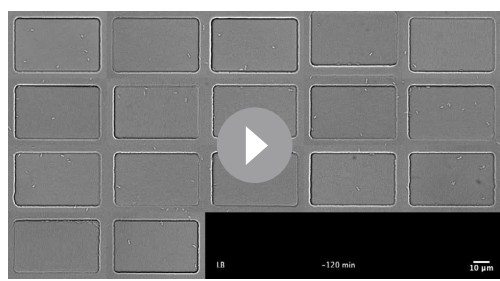

**Video 14.** Time-lapse movie of 17 representative persister cells in the post-late stationary phase cell populations of MF1 against the exposure to 200 µg/mL Amp for 3.5 h in LB. This movie shows 17 representative persister cells observed in one of the single-cell measurements of the post-late stationary phase cells of MF1 exposed to 200 µg/mL Amp for 3.5 h in LB (*Figure 3—figure supplement 1*). We used a 40× objective in this measurement. Scale bar, 10 µm.
https://elifesciences.org/articles/79517/figures#video14

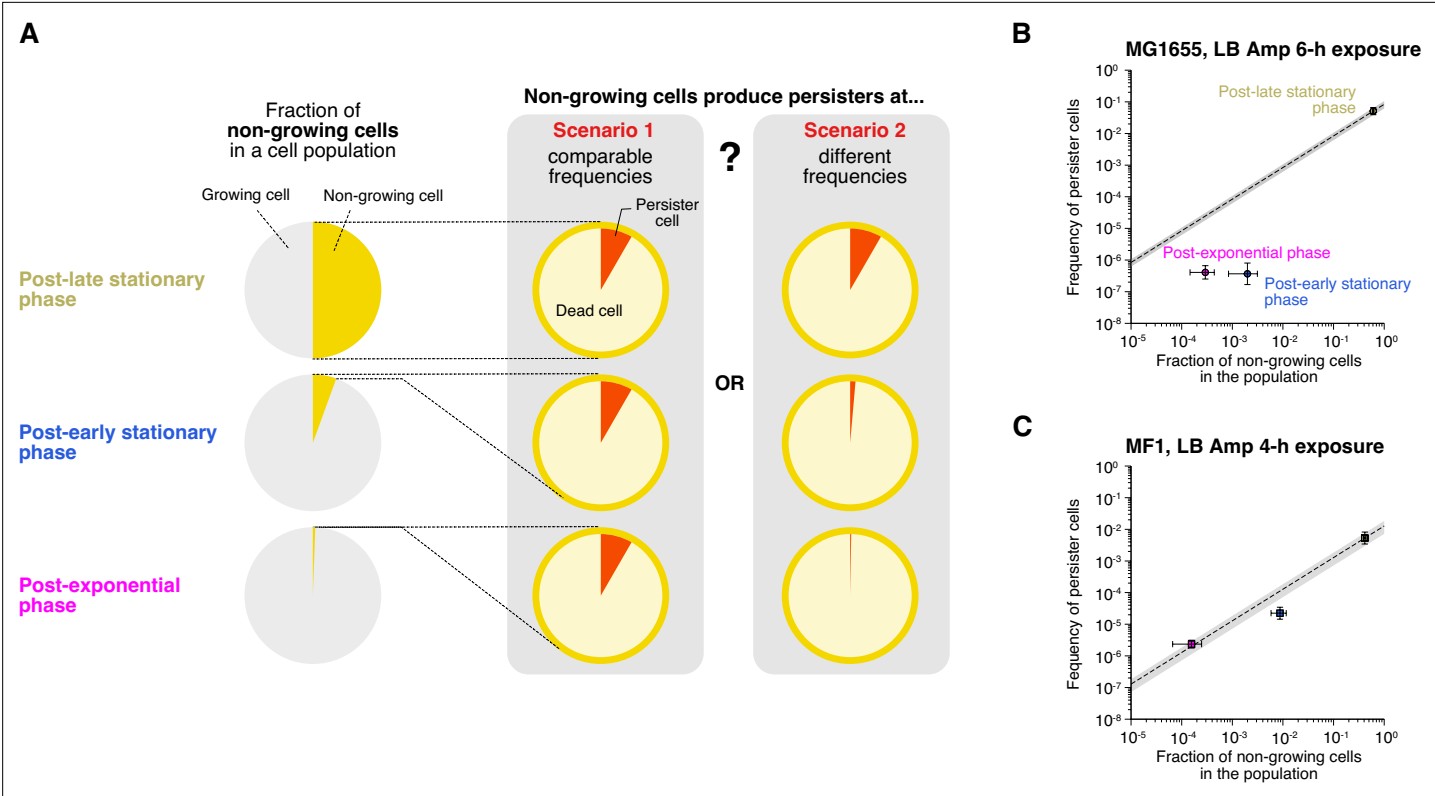

**Figure 7.** Characterization of growth phase-dependent changes in the proportion of non-growing cells in cell populations and their survivability against Amp exposure. (**A**) Schematic illustration of the purpose of the analysis. In this analysis, we asked how the fraction of non-growing cells in a cell population changes depending on growth phase and whether these non-growing cells produce persisters (non-growing persisters) at comparable frequencies (scenario 1) or at different frequencies (scenario 2) under different conditions. (**B**) Relationship between the fractions of non-growing cells and the frequencies of surviving cells of MG1655 after exposure to 200 μg/mL Amp for 6 h in LB medium. The horizontal axis represents the fraction of non-growing cells characterized by single-cell measurements. The vertical axis represents the frequency of surviving cells estimated from the data points at 6 h in the population killing curves shown in *Figure 1—figure supplement 3A*, *Figure 6B* (points). The error bars along the horizontal axis represent the binomial standard errors, and those along the vertical axis represent the standard errors of the frequencies of the surviving cells. The dashed line represents the predicted frequency of non-growing persisters in a cell population, i.e., *Equation 1* with the assumption of constant $s$ (scenario 1). The gray background indicates the prediction error range derived from the estimation error range of $s$ ($6.18 \times 10^{-2}$–$1.06 \times 10^{-1}$). (**C**) Relationship between the fractions of non-growing cells and the frequencies of surviving cells of MF1 after exposure to 200 μg/mL Amp for 4 h in LB medium. The frequency of surviving cells was estimated from the data points at 4 h in the population killing curves in *Figure 1—figure supplement 3A*, *Figure 6—figure supplement 1C*. The gray background indicates the prediction error range derived from the estimation error range of $s$ ($7.19 \times 10^{-3}$–$1.84 \times 10^{-2}$).

of non-growing cells in the populations increased significantly with the time spent under stationary phase conditions.

We next asked whether these non-growing cell fractions with different cultivation histories have persister cells at comparable frequencies (*Figure 7A*). If the probability of non-growing cells being persisters ($s$) is independent of the conditions, the frequency of non-growing persisters in the population ($f$) should follow

$$f = sP_n \qquad (1)$$

where $P_n$ is the fraction of non-growing cells in the population immediately before the drug exposure. Since all persister cells in the post-late stationary phase cell populations for which we identified their pre-exposure growth trait were non-growing persisters (*Figure 3A*), we estimated $s$ as $f^{(late)}/P_n^{(late)} = (8.4 \pm 2.2) \times 10^{-2}$ for MG1655, where $f^{(late)}$ is the frequency of persister cells in the post-late stationary phase cell populations estimated from the killing curve at 6 h under Amp exposure (*Figure 6B*), and $P_n^{(late)}$ is the frequency of non-growing cells in the post-late stationary phase cell populations estimated above.

**Video 15.** Viability check of the non-regrowing non-growing cells by PI staining (6-h Amp exposure). This movie shows the time-lapse observation of the post-late stationary phase cells of MG1655/pUA66-P$_{rpsL}$-*gfp* in LB medium. We used a 100× objective in this measurement. Left: Merged images of the bright-field (grayscale), GFP (green), and PI (red) channels; Middle: GFP channel; Right: PI channel. PI was added to the flowing media from the beginning of the Amp exposure as indicated by the text at the bottom of the movie. We found many non-growing cells that did not regrow even after the removal of Amp. They were visible in the GFP channel but negative for PI staining. Scale bar, 10 μm.

https://elifesciences.org/articles/79517/figures#video15

In *Figure 7B*, the diagonal line shows the frequency of non-growing persister cells at different non-growing cell frequencies ($P_n$) predicted by *Equation 1*. We also show in *Figure 7B* the frequencies of persister cells (non-growing persisters + growing persisters) in the post-exponential phase and post-early stationary phase cell populations estimated from their killing curves (*Figure 6B*). The result shows that the frequencies of persister cells under these conditions are lower than the predicted frequencies of non-growing persisters. Since the persisters in the post-early stationary phase cell populations contained growing persisters, albeit at a low frequency, and all the persisters detected in the post-exponential phase cell populations were growing persisters (*Figure 3A*), the true frequencies of non-growing persisters under these conditions should be lower than the points in *Figure 7B*, especially for the post-exponential phase condition. Therefore, the non-growing cells in the post-exponential phase and post-early stationary phase cell populations are less likely to be non-growing persisters than those in the late-stationary phase cell populations.

We performed the same analysis with MF1, referring to the frequencies of surviving cells of the killing curves at 4 h under Amp exposure (*Figure 6—figure supplement 1C*) and to the frequencies of non-growing cells in the cell populations estimated by single-cell time-lapse observation. We used the frequencies of surviving cells at 4 h instead of 6 h, considering that the decay rate shift of the killing curves of MF1 occurs earlier than that of MG1655 (*Figure 6B* and *Figure 6—figure supplement 1C*). The probability of non-growing cells being persisters was estimated to be $s = f^{(late)}/P_n^{(late)}$ = (1.3±0.6)×10$^{-2}$. The measured frequency of persister cells in the post-early stationary phase cell populations was slightly lower than the predicted frequency of non-growing persisters (*Figure 7C*). On the other hand, the measured frequency of persister cells in the post-exponential phase cell populations was approximately equal to the predicted frequency of non-growing persisters in this condition (*Figure 7C*). However, since the majority of persister cells in the post-exponential phase cell populations were growing persisters (*Figure 3—figure supplement 1*), the actual frequency of non-growing persisters in this condition should be lower than the predicted frequency. Therefore, similar to MG1655, non-growing cells in the post-exponential and post-early stationary phase cell populations of MF1 are less likely to be persisters than those in the post-late stationary phase cell populations.

Taken together, these analyses suggest that prolonged incubation under stationary phase conditions increases the persister fraction of non-growing cells. This conclusion is consistent with a previous study by Luidalepp, et al. in which the number of non-growing cells was estimated by the number of non-lysed cells in a batch culture incubated for 3 h in Amp-containing fresh media after transfer from cultures of varying lengths of stationary phase (*Luidalepp et al., 2011*).

## Growing persisters are predominant against ciprofloxacin in both post-exponential and post-late stationary phase cell populations

Having observed that the persisters against Amp in the post-exponential phase cell populations of MG1655 and MF1 were predominantly growing persisters (*Figure 3*, *Figure 3—figure supplement 1*), we next investigated persistence against another antibiotic drug, ciprofloxacin (CPFX). While Amp is a $\beta$-lactam antibiotic targeting cell walls, CPFX is a fluoroquinolone antibiotic targeting DNA gyrase. As shown previously in *Figure 1—figure supplement 3B*, post-exponential phase cell populations of MG1655 and MF1 in M9 minimal medium exposed to 1 μg/mL CPFX (32×MIC, *Figure 1—figure supplement 2B*) exhibited persistent killing curves, confirming the presence of persistence against

CPFX. The frequencies of persisters evaluated at the 6 h time point were $4.5 \times 10^{-5}$ ($3.2 \times 10^{-5}$–$6.4 \times 10^{-5}$) for MG1655 and $1.4 \times 10^{-4}$ ($8.9 \times 10^{-5}$–$2.2 \times 10^{-4}$) for MF1 (*Figure 1—figure supplement 3B*).

To characterize the dynamics of individual persister cells in the populations, we conducted time-lapse experiments using the MCMA device (*Figure 8A–D*, *Figure 8—figure supplement 1*, *Videos 16 and 17*). We loaded exponentially growing cells in M9 medium into the MCMA device, allowed them to continue growing in the device by flowing fresh medium for 4 h, exposed them to 1 µg/mL CPFX for 6 hr, and monitored the re-proliferation of surviving cells (*Figure 1C and D*). We detected 32 persister cells that regrew after CPFX exposure among $2.0 \times 10^5$ drug-exposed cells for MG1655 (*Figure 3*) and 148 persister cells among $8.3 \times 10^5$ drug-exposed cells for MF1 (*Figure 3—figure supplement 1*). We found that all persister cells for which we were able to identify the pre-exposure growth trait were growing persisters (*Figure 3*, *Figure 3—figure supplement 1*, *Videos 16 and 17*). Therefore, as observed for the Amp exposure, growing persisters were predominant in the post-exponential phase cell populations exposed to CPFX.

We also analyzed persister cells of MG1655 against CPFX in the post-late stationary phase cell populations in M9. Unlike the result for Amp, the post-late stationary phase cell populations did not produce significantly more persisters against CPFX than the post-exponential phase cell populations in batch cultures (*Figure 8—figure supplement 2*). We loaded cells in the late stationary phase (24 h time point in the growth curve in *Figure 1—figure supplement 4B*) into the MCMA device and allowed the cells to regrow from the stationary phase condition by changing the media around the cells from the stationary phase conditioned medium to fresh M9 medium at the beginning of the time-lapse measurement (*Figure 1C and D*). After 4 h of pre-exposure incubation in the fresh medium, we exposed the cells to 1 µg/mL CPFX for 6 h. We then allowed them to regrow by switching the flowing medium back to the drug-free fresh medium and monitored the re-proliferation.

We detected 218 persister cells of MG1655 that re-proliferated upon the removal of CPFX and found that all persister cells for which we could identify the pre-exposure growth trait (170 of the 218 persister cells) were growing persisters (*Figure 3* and *Video 18*). Therefore, growing persisters were dominant in both post-exponential and post-late stationary phase cell populations exposed to CPFX in M9 medium. Notably, a previous study (*Goormaghtigh and Van Melderen, 2019*) detected only non-growing persister cells against ofloxacin when cells were sampled from the stationary phase and allowed to grow in a microfluidic device for 7 h in the flow of a fresh medium (MOPS medium) before drug exposure. Our results show that the presence of growing persisters is not limited to the cell populations placed under well-controlled exponential growth conditions for prolonged periods, depending on the antibiotic type.

We also analyzed in detail the growth and division dynamics of the persister cells in the post-exponential phase cell populations during the pre-exposure, exposure, and post-exposure periods (*Figure 8A–D*, *Figure 8—figure supplement 1*, *Videos 16 and 17*). Unlike Amp, the CPFX exposure did not lyse cells but arrested their growth and division, including those of the persister cells (*Figure 8A–D*, *Figure 8—figure supplement 1*, *Videos 16 and 17*). After the removal of CPFX, the persister cells resumed growth and many of them became filamentous (*Figure 8A–D*, *Figure 8—figure supplement 1*, *Videos 16 and 17*). The filamentous cells restarted cell divisions and produced normal-sized, and normally growing and dividing, progeny cells. The observed post-exposure filamentation dynamics of persister cells are consistent with what was reported for *E. coli* persisters against another fluoroquinolone antibiotic, ofloxacin (*Goormaghtigh and Van Melderen, 2019*). On the other hand, we found another type of persister cells that resumed normal growth and division without showing prominent filamentation (*Figure 8C and D*, *Figure 8—figure supplement 1*). Therefore, extensive filamentation after CPFX treatment is not necessarily required to restore normal growth and division in persister cells.

In addition, comparison of the pre-exposure division rate showed no significant difference between growing persisters of MG1655 (0.94±0.06 doublings/h) and non-surviving cells (0.86±0.03 doublings/h; $p = 0.12$, Wilcoxon rank sum test) (*Figure 8E*). Thus, MG1655 persisters to CPFX grow as fast as non-surviving cells prior to drug exposure. This result is again consistent with the results observed for the persistence of post-exponential phase *E. coli* cell populations against ofloxacin (*Goormaghtigh and Van Melderen, 2019*).

Microbiology and Infectious Disease | Physics of Living Systems

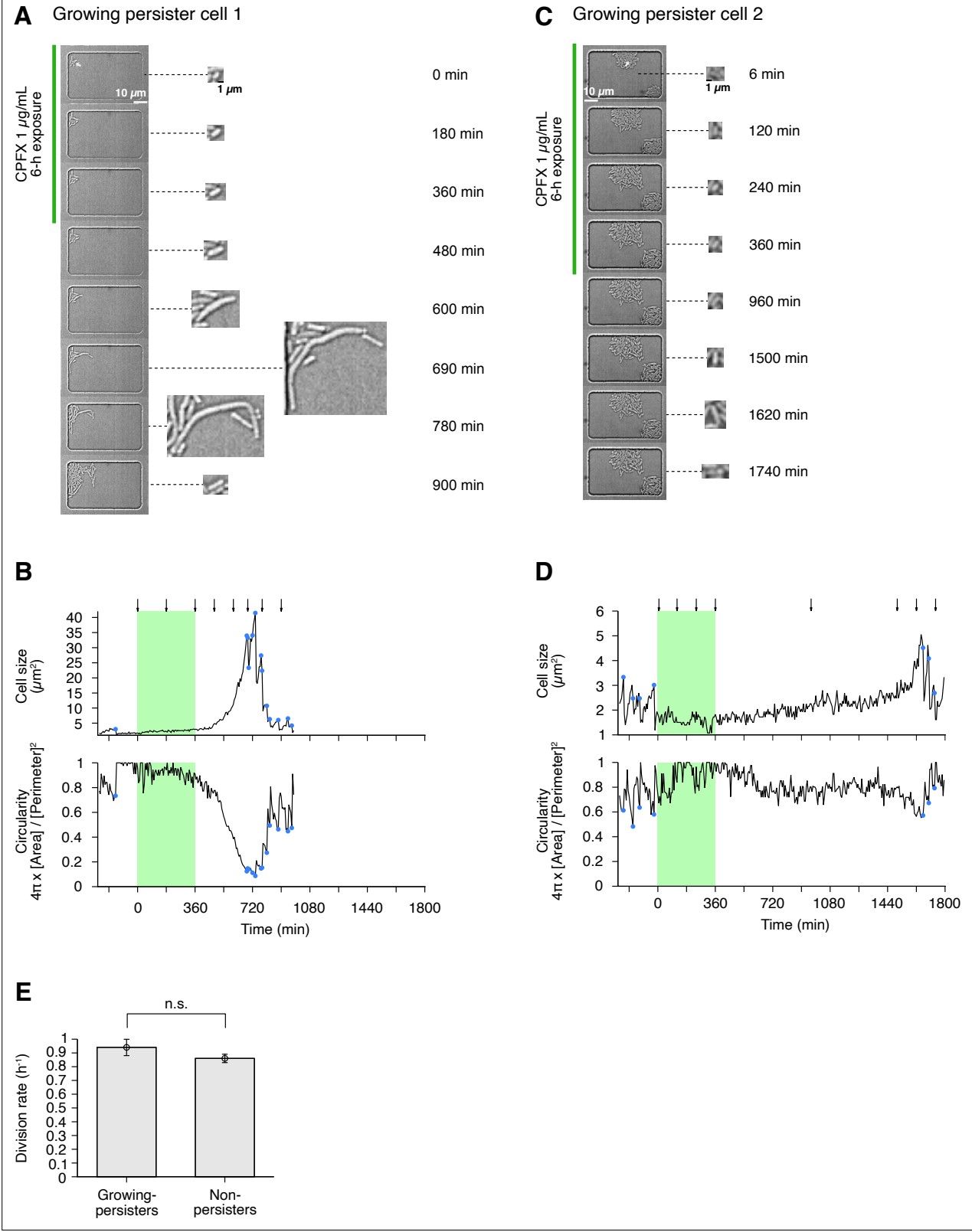

**Figure 8.** Persisters against CPFX observed in the post-exponential phase cell populations of MG1655 in M9. (**A**) Time-lapse images of a growing persister showing extreme filamentation after exposure to 1 µg/mL CPFX for 6 h. The left image sequences show the bright-field images. The right image sequences show the enlarged micrographs of the persister cell. The green line to the left of the images indicates the images under the CPFX exposure. The white arrow indicates the growing persister cell. (**B**) Temporal changes in cell size (top graph) and cell shape circularity (bottom graph) of

20 of 33

*Figure 8 continued*

the growing persister shown in A. The green background indicates the 6-h period of the CPFX exposure. The black arrows indicate the time points at which the micrographs in A were taken. The blue dots indicate the points where cell divisions occurred. (**C**) Time-lapse images of a growing persister showing no extreme filamentation during and after the CPFX exposure. Images are displayed in the same way as in A. (**D**) Temporal changes in cell size (top graph) and cell shape circularity (bottom graph) of the growing persister shown in C. (**E**) Comparison of pre-exposure division rates between growing persisters ($n$ = 24) and non-surviving cells ($n$ = 76). Due to the difficulty in counting the number of pre-exposure cell divisions, five growing persisters were excluded from this analysis. No significant difference in division rates was detected at the significance level of 0.05 ($p$ = 0.12, Wilcoxon rank sum test).

The online version of this article includes the following figure supplement(s) for figure 8:

**Figure supplement 1.** Single-cell dynamics of MG1655 persisters against CPFX in the post-exponential phase cell populations in M9.

**Figure supplement 2.** Killing curves of post-late stationary phase cell populations of MG1655 against CPFX.

## Discussion

Like genetically mutated resistant or tolerant bacteria, phenotypic variants of bacteria that arise in isogenic cell populations and cause persistence also pose a threat to global human health by reducing the efficacy of antibiotic treatment. Furthermore, the survival of persister cells under prolonged anti-biotic treatment provides time for bacteria to develop genetic mutations that lead to irreversible antibiotic resistance. Therefore, understanding how individual persister cells evade the deleterious effects of antibiotics is critical to counteracting persister cells. However, due to the extremely low frequency of persister cells in cellular populations, direct visualization of their dynamics is still limited to a handful of cases.

Time-lapse single-cell observations with the MCMA microfluidic device enable direct visualization of persister cells existing at extremely low frequencies in isogenic bacterial cell populations. Single-cell measurements with this device revealed that actively growing cells produced persisters predominantly when Amp was applied to post-exponential phase cell populations (*Figure 3*). Frequencies of non-growing persisters increased in the cellular populations sampled from the early and late stationary phases in LB (*Figure 3*). Furthermore, we detected only growing persisters when CPFX was applied to post-exponential and post-late stationary phase cell populations in M9 (*Figure 3*). These results show that, as previously suggested in *Zou et al., 2021*, multiple survival modes are available for *E. coli* cells to achieve persistence not only against different antibiotics but even against the same drug. Furthermore, distinct survival modes can coexist within a single persistent cell population (*Figure 3*, *Figure 3—figure supplement 1*). In such cases, population-based analyses on bacterial persistence with an assumption of a single underlying mechanism would provide blurred results and bias our understanding of the phenomenon.

Despite the relatively high medium exchange rate in the MCMA device, we cannot exclude the possibility that a small amount of antibiotic may remain in the device, for example due to non-specific adsorption on the internal surface of the

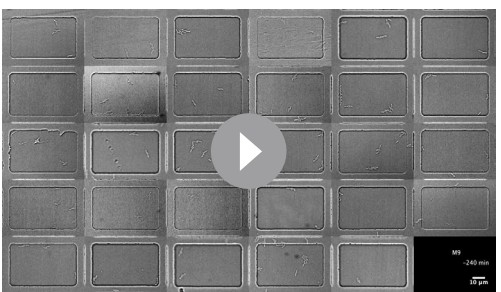

**Video 16.** Time-lapse observation of 29 representative persister cells in the post-exponential phase cell populations of MG1655 exposed to 1 µg/mL CPFX for 6 h in M9. This movie shows 29 representative persister cells observed in one of the single-cell measurements of the post-exponential phase cells of MG1655 exposed to 1 µg/mL CPFX for 6 h in M9 (*Figure 3*). We used a 40× objective in this measurement. Scale bar, 10 µm.

https://elifesciences.org/articles/79517/figures#video16

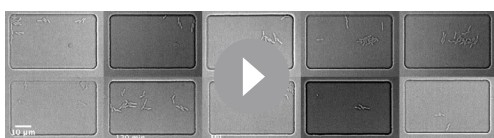

**Video 17.** Time-lapse observation of 10 representative persister cells in the post-exponential phase cell populations of MF1 exposed to 1 µg/mL CPFX for 6 h in M9. This movie shows 10 representative persister cells observed in one of the single-cell measurements of the post-exponential phase cells of MF1 exposed to 1 µg/mL CPFX for 6 h in M9 (*Figure 3—figure supplement 1*). We used a 40× objective in this measurement. Scale bar, 10 µm.

https://elifesciences.org/articles/79517/figures#video17

**Video 18.** Time-lapse observation of 10 representative persister cells in the post-late stationary phase cell populations of MG1655 exposed to 1 μg/mL CPFX for 6 h in M9. This movie shows 10 representative persister cells observed in one of the single-cell measurements of the post-late stationary phase cells of MG1655 exposed to 1 μg/mL CPFX for 6 h in M9 (*Figure 3*). We used a 40× objective in this measurement. Scale bar, 10 μm.

https://elifesciences.org/articles/79517/figures#video18

microchambers. The residual antibiotics may influence the physiological state of the cells and the regrowth kinetics in the post-exposure periods, even if they are present at concentrations significantly lower than the MICs (*Davies et al., 2006*; *Andersson and Hughes, 2014*). However, the frequencies of persister cells in the cell populations in our single-cell measurements are comparable to those in the batch culture measurements. Therefore, the removal of antibiotics in our device is at least as efficient as in the batch culture assay.

It is of note that the presence of growing persisters was not limited to the cell populations placed under prolonged exponential growth conditions, which are rarely found in natural contexts. In the CPFX treatment, all persister cells for which we identified their pre-exposure growth traits were also growing persisters in the post-late stationary phase conditions (*Figure 3*). This result contrasts with what has been reported for persistence to ofloxacin, where only non-growing persisters were detected when ofloxacin was exposed to the post-stationary phase cell populations after 7 h of pre-exposure incubation in fresh medium (*Goormaghtigh and Van Melderen, 2019*). The experimental conditions such as antibiotics (ofloxacin or CPFX), culture media (MOPS medium or M9 medium), and microfluidic device types were different between these studies, which may have caused the difference in the dominant survival modes. Although the reasons for the discrepancy in observations against similar fluoroquinolone antibiotics remain elusive, our result suggests that growing persisters may be relevant to the survival of cell populations under a wide range of conditions, including natural and clinical settings.

The coexistence of multiple survival modes within a single cell population may be related to the killing curves fitting better to a triphasic exponential decay than to a biphasic exponential decay under several conditions (*Figure 1—figure supplement 3*, *Figure 6B*, *Figure 6—figure supplement 1* and *Figure 8—figure supplement 2*). In general, two qualitatively different scenarios can generate such multiphasic killing curves: (1) the presence of multiple types of persister cells in the cell populations and (2) the antibiotic-induced increase in refractoriness of surviving cells during treatment. The observation of killing curves with more than two killing phases is not limited to our study. Svenningsen et al. have reported multiphasic killing curves for *E. coli* exposed to CPFX for 7 days and have suggested the contributions of different intracellular programs to the survival in different killing phases (*Svenningsen et al., 2022*). Although the two scenarios mentioned above are not mutually exclusive, elucidating the contribution of these qualitatively different processes would be crucial for identifying the key events underlying the persistence of cell populations against antibiotics.

Interestingly, some of the growing persisters against Amp exhibited radical morphological changes and transitioned to the cell shapes that resemble L-form or spheroplast cells (*Zou et al., 2021*; *Figures 2A–D, 4C and D*). L-form bacteria are deficient in cell walls but can sustain growth and division under osmoprotective conditions (*Allan et al., 2009*; *Errington et al., 2016*; *Chikada et al., 2021*). L-form cells can be produced by treating bacteria with exogenous lytic enzymes or β-lactam antibiotics under osmoprotective conditions (*Allan et al., 2009*; *Chikada et al., 2021*). It is known that L-form bacteria are resistant to most antibiotics targeting cell wall synthesis, particularly β-lactams (*Allan et al., 2009*; *Errington et al., 2016*). Therefore, if the L-form-like cells produced by the Amp exposure could manage to protect themselves from osmotic pressure, they might be able to persist for an extended period under the Amp exposure. Importantly, the coexistence of non-walled spherical persisters and non-spherical rod-shaped persisters has also been reported for *Acinetobacter baumannii* cells treated with a β-lactam antibiotic, meropenem (*Zou et al., 2020*). Therefore, the emergence and survival of L-form or spheroplast-like persisters may be widespread in the persistence to β-lactam antibiotics. Since L-form cells are often found as antibiotic-resistant organisms and isolated from humans and animals in infectious diseases (*Allan et al., 2009*), this survival mode might be relevant to clinical settings. However, we also remark that such L-form-like cells did

not appear upon CPFX exposure. Therefore, the transitions to L-form-like cells might be the general characteristics of growing persisters against Amp and other $\beta$-lactam antibiotics, but these would not apply to the other non-$\beta$-lactam antibiotics.

Relatedly, Männik et al. reported the emergence of *E. coli* cells with aberrant cell shapes when cells are forced to pass through channels whose height or width is smaller than the cell width (***Männik et al., 2009***). Similar to the L-form-like cells observed in our measurements, these aberrant cells are also able to grow and divide and restore the rod shape after 1–2 days. Thus, the transition to aberrant cell shapes can be induced both physically and chemically, but *E. coli* cells can maintain not only viability but also proliferative capacity against such various perturbations. This resilience to changes in cellular morphology may have contributed in part to the survival of growing persisters against Amp. In the MCMA device, *E. coli* cells were enclosed in chambers with a depth comparable to the cell width by a semipermeable cellulose membrane. Therefore, although Amp exposure would be the direct trigger of the morphological changes, the spatial confinement of the chambers may also have promoted the cell shape transition.

The result that the incubation under stationary phase conditions enhanced the survivability of non-growing cells against Amp (***Figure 7***) indicates that the states of non-growing cells were not identical across the growth phases, as suggested previously in ***Levin et al., 2014***; ***Pu et al., 2016***. It is noteworthy that in our experiments, a different dominant survival mode of persister cells was observed for the post-early and post-late stationary phase cell populations than for the post-exponential phase cell populations, despite the 2 h pre-exposure cultivation in fresh LB medium. Therefore, 2 h of incubation in fresh medium is not sufficient to erase the memory of the previous stationary phase conditions. Relatedly, the dominant survival modes were different between post-exponential and post-early stationary phase conditions despite the comparable frequency of persister cells in the cell populations (***Figure 6B***). This suggests that the susceptibility of actively growing cells to Amp also depends on the previous cultivation condition; actively growing cells in the post-exponential phase cell populations are more likely to produce persister cells than those in the post-early stationary phase cell populations.

The role of RpoS, a specialized sigma factor that controls the general stress response, in antibiotic persistence has been debated and no consistent conclusion has been reached (***Hong et al., 2012***; ***Wu et al., 2015***; ***Valencia et al., 2024***). Comparing the results between the wildtype strain MG1655 and the RpoS-defective strain MF1 may provide insight into this issue. Our results suggest that RpoS affects the frequency of persister cells (***Figure 3***, ***Figure 3—figure supplement 1***) as well as the population dynamics in the presence of antibiotics (***Figure 1—figure supplement 3***, ***Figure 6—figure supplement 1A and B***). The effect of retaining intact RpoS can be both positive and negative depending on the culture conditions and antibiotics. For example, in the post-late stationary phase cell populations in LB and post-exponential phase in M9, the RpoS-deficient MF1 strain consistently contained fewer surviving cells than the MG1655 strain (***Figure 1—figure supplement 3C***, and ***Figure 6—figure supplement 1B***). However, under the other conditions examined in this report, the frequency of surviving cells was higher in MF1 than in MG1655 at the end of prolonged drug exposure (***Figure 1—figure supplement 3A and B***, and ***Figure 6—figure supplement 1A***). These results confirm the differential importance of intact RpoS function and again suggest that multiple factors contribute to antibiotic persistence.

As demonstrated here, our MCMA device can detect the persister cells existing at frequencies of $10^{-5}$ within a cellular population. This is the order at which one can detect persisters in most combinations of bacterial species and antibiotics, including clinically important ones (***Spoering and Lewis, 2001***; ***Johnson and Levin, 2013***; ***Gomez and McKinney, 2004***). Although our analysis is limited to two types of antibiotics, multiple survival modes have been characterized in *E. coli* persistence. Therefore, it would be crucial to unravel how individual cells of each bacterial species persist to different types of antibiotics under different conditions. Direct single-cell observations would unmask the diversity of persistence mechanisms and might enable rational designs of antibiotic treatments against bacterial infectious diseases.

## Methods

### *E. coli* strains and culture conditions

The *E. coli* strains MG1655 and MF1 (MG1655 *rpoS-mcherry*/pUA66-P$_{rpsL}$-*gfp*) were mainly used in this study. pUA66-P$_{rpsL}$-*gfp* is a low-copy plasmid taken from the *E. coli* promoter library (*Zaslaver et al., 2006*). In the propidium iodide (PI) staining experiment, we used the MG1655/pUA66-P$_{rpsL}$-*gfp* strain to examine the intactness of cellular states by GFP fluorescence in addition to susceptibility to PI staining. In the experiment testing the susceptibility of the MF1 strain to oxidative stress, we also used the MG1655 Δ*rpoS* strain.

For constructing the MF1 strain, a DNA fragment containing *mcherry* and kanamycin-resistance genes was amplified by PCR with the template pLmCherryK6Δterm and the primers rpoS-mcherry-F (CAAACGCAGG GGCTGAATAT CGAAGCGCTG TTCCGCGAGA GTGATTTTAT GGTGAGCAAG GGCGAGGA) and rpoS-mcherry-R (CAGCCTCGCT TGAGACTGGC CTTTCTGACA GATGCTTACA TTCCGGGGAT CCGTCGACC) and integrated downstream of the *rpoS* locus on the chromosome of BW25113 strain by λ-Red recombination (*Datsenko and Wanner, 2000*) to express the RpoS-mCherry fusion protein from the native locus. The *rpoS-mcherry* gene and kanamycin-resistance gene flanked with two FRT sequences were transferred to MG1655 by P1 transduction. Kanamycin-resistance gene was removed by transforming pCP20 plasmid. The pCP20 plasmid was removed from the host cells by culturing them at 37°C overnight.

To construct the MG1655 Δ*rpoS* strain, the FRT-kanamycin resistance gene-FRT cassette in the JW5437 strain (BW25113 Δ*rpoS*, a strain in the Keio collection [*Baba et al., 2006*]) was transferred into MG1655 by P1 transduction. Removal of the kanamycin resistance gene and subsequent exclusion of the pCP20 plasmid from the cell was performed as for the construction of the MF1 strain.

We cultured the cells with either Luria-Bertani (LB) broth (Difco) or M9 minimal medium (Difco) supplemented with 1/2 MEM amino acids solution (SIGMA) and 0.2% (w/v) glucose. All cultivation was done at 37°C.

### Growth curve measurements

5 μL of the glycerol stock cell suspension was inoculated into 2 mL of LB or M9 minimal medium containing 40 μg/mL of kanamycin (Kan), and we cultured the cells at 37°C for 15 h (LB) or 16 h (M9) with shaking. Kan was added to the culture media to avoid the loss of the GFP-expressing plasmid (pUA66-P$_{rpsL}$-*gfp*) during the pre-culture of MF1. We diluted this pre-culture cell suspension with fresh LB or M9 medium (without Kan) to the cell density corresponding to the optical density at 600 nm (OD$_{600}$) being 0.005 (LB) or 0.01 (M9). The total volume of the diluted cell cultures was adjusted to 50 mL. After the dilution, we started culturing cells at 37°C with shaking. We sampled 100 μL of the cell cultures every 1 h and measured OD$_{600}$ with a spectrophotometer (UV-1800, Shimadzu). We repeated the measurements four times for MG1655 in LB, three times for MG1655 in M9, five times for MF1 in LB, and five times for MF1 in M9. The mean and standard error of the logarithm of the OD$_{600}$ were calculated for each sampling time point and plotted as the point and error bar in *Figure 1—figure supplement 4*. In *Figure 1—figure supplement 1A and B*, we show only the growth curves from one experiment for each condition, in which we measured RpoS-mCherry fluorescence of individual cells in addition to the optical density of the cell cultures.

### Killing curve assay

An overview of the sample preparation for the killing curve assay is shown schematically in *Figure 6—figure supplement 1*. To measure killing curves in batch cultures, we first prepared 50 mL cell suspensions of OD$_{600}$ = 0.005 (LB) or OD$_{600}$ = 0.01 (M9) following the same protocol as for growth curve measurements (first pre-culture). We incubated the diluted cell suspensions at 37°C with shaking (second pre-culture) and sampled a portion of the cell cultures at a pre-determined time point (2 h for the exponential phase in LB; 8 h for the early stationary phase in LB; 24 h for the late stationary phase in LB; 4 h for the exponential phase in M9; and 24 h for the late stationary phase in M9). To match the pre-culturing conditions between the killing curve assays and the single-cell measurements, we diluted the extracted cell suspension with the fresh LB or M9 medium to the cell density with which the OD$_{600}$ of the diluted cell suspension would reach 0.05 in the pre-culture of 2 h (LB) or 4 h (M9). The volume of the cell cultures was adjusted to 50 mL. After the pre-culture, we added the antibiotics (Amp or CPFX) to the cell culture and incubated it at 37°C with shaking. The concentrations of the

antibiotics were adjusted to 200 µg/mL for Amp and 1 µg/mL for CPFX. We extracted 50 µL or 500 µL of the cell suspensions at the pre-determined time points to estimate the number of viable cells in the cellular populations.

To avoid underestimating the viable cells due to non-culturability on solid agar (*Gelman et al., 2012*; *Nosho et al., 2018*), we estimated the viable cell number by a limiting dilution method. We conducted serial dilution of the extracted cell suspension (10× dilution in each dilution step) in 5 mL. We sampled 200 µL of each diluted cell suspension into a well in a 96-well plate. For each dilution condition, cell suspensions were mostly dispensed into 12 wells. For later time points of the post-exponential and post-early stationary phase cell populations, cell suspensions were dispensed into 21, 24, or 36 wells to detect low-frequency surviving cells. We incubated the 96-well plates at 37°C for 36 h (LB) or 48 h (M9), and the turbidity of each well was inspected.

We found one or two dilution conditions for each sample taken from different time points where both turbid and non-turbid wells coexist among the dispensed 12 or more wells. Assuming Poissonian allocation of viable cells, the proportion of non-turbid wells, $x_0$, is equal to $e^{-dyN_v}$, where $N_v$ is the original concentration of viable cells, $d$ is the dilution level, and $y$ (= 0.2 mL) is the volume; then, $N_v = (yd)^{-1} \ln(1/x_0)$. Thus, we estimated the concentration of viable cells in the cell population as $N_v = (5/d) \ln(W/k)$ cells/mL, where $W$ is the number of dispenced total wells, and $k$ is the number of non-turbid wells. For example, if we found three non-turbid and nine turbid wells for the cell suspension with the dilution of $10^{-6}$, we could estimate the viable cell density as $N_v = 5 \times 10^6 \times \ln(12/3)$ cells/mL $= 7 \times 10^6$ cells/mL. When the coexistence of turbid and non-turbid wells was found in the two dilution conditions, we estimated the viable cell density of this time point by the geometric mean of the two estimated values.

We repeated the assay three or more times for each culture condition. The mean and standard error of the logarithm of the estimated viable cell density for each sampling time were calculated and shown as the point and error bar in *Figure 1—figure supplement 3*, *Figure 6B*, *Figure 6—figure supplement 1*, and *Figure 8—figure supplement 2*.

## MIC measurements

5 µL of the glycerol stock of the *E. coli* strain was inoculated into 2 mL of LB or M9 medium and incubated at 37°C with shaking for 15 h (LB) or 16 h (M9). We added Kan at the concentration of 40 µg/mL when we cultured the MF1 strain from the glycerol stock to avoid the loss of the GFP-expressing plasmid (pUA66-P$_{rpsL}$-*gfp*) during the pre-culture. The cell cultures were diluted with a fresh LB or M9 medium to the cell density corresponding to OD$_{600}$ = 0.005 (LB) or OD$_{600}$ = 0.01 (M9). The diluted cell cultures of 2 mL were incubated at 37°C with shaking for 2.5 h (LB) or 3 h (M9) to let the cell cultures enter exponential phases. The cell cultures were again diluted with a fresh medium to the cell density corresponding to OD$_{600}$ = 0.0002. We mixed 100 µL of the diluted cell suspension with 100 µL of the medium containing Amp or CPFX and dispensed it into a well of the 96-well plate. We prepared 12 antibiotic concentration conditions with 2-fold serial differences (Amp: $2^{-2}$–$2^8$ µg/mL (+0 µg/mL); CPFX: $2^{-10}$–$2^0$ µg/mL (+0 µg/mL)). We used three wells for each drug concentration condition. The cell cultures in the 96-well plates were incubated at 37°C with shaking for 23 h. We measured the optical density at 595 nm (OD$_{595}$) of each well using a plate reader (FilterMax F5, Molecular Devices) and calculated the geometric mean of the three wells as the optical density of the cell culture in each drug concentration. We repeated the experiment three times and determined the MIC as the drug concentration above which the geometric mean of OD$_{595}$ among the three replicate experiments were below 0.001.

## Measurement of RpoS-mCherry fluorescence level transition along growth curves

We followed the same protocol as in 'Growth curve measurements' above to prepare cell cultures of MF1 for measuring the growth curves and RpoS-mCherry fluorescence level transitions shown in *Figure 1—figure supplement 1A and B*. In addition to measuring OD$_{600}$, we collected 1 µL of cell suspension every hour and placed it between a PBS block solidified with 1.5% agarose and a coverslip.

We acquired brightfield, RpoS-mCherry fluorescence, and GFP fluorescence images using a microscope. The optical configuration and settings of the microscope used for this measurement were the same as those used for the time-lapse measurement described in detail below (see 'Time-lapse

observation with the MCMA microfluidic device'). We used Fiji (*Schindelin et al., 2012*) to find the outlines of individual cells and quantify their RpoS-mCherry fluorescence levels.

## Susceptibility test against $H_2O_2$

We used the MG1655, MG1655 Δ*rpoS*, and MF1 strains in this test. 5 µL of the glycerol stock of each strain was inoculated into 2 mL of M9 medium and incubated overnight at 37°C with shaking. Kan was added to the culture of MF1 during this pre-culture to avoid the loss of the GFP-expressing plasmid (pUA66-P$_{rpsL}$-*gfp*). The optical density of each overnight culture was measured, and the culture was diluted in 20 mL of fresh M9 medium to a cell density corresponding to $OD_{600}$ = 0.01. The culture was incubated for 4 h with shaking. 2 mL of cell suspension was taken into each of two microtubes per strain, and the cells were spun down by centrifugation. The cell pellet was resuspended in 2 mL of sterile water, and the cell suspension was transferred to a glass test tube. To one of the tubes, 4 µL of 30% $H_2O_2$ was added to adjust the final concentration of $H_2O_2$ to 20 mM. An equal volume of sterile water was added to the other tube (control). These cell cultures were incubated at 37°C for 20 min with shaking. 500 µL of cell suspension was transferred from each test tube to a microtube. The cells were centrifuged, and the cell pellet was resuspended in 500 µL of M9 medium. The cell suspension was serially diluted in multiple steps with a 5-fold dilution in each step. 100 µL of each cell suspension was plated on an LB agar plate and incubated overnight at 37°C.

By counting the number of colonies on the plate for the dilution condition with 30–300 colonies, we estimated the viable cell density in each cell culture. When we found 30–300 colonies for multiple dilution conditions, we adopted the mean of them as the estimated viable cell density. For each strain, the viable cell density of the cell culture exposed to 20 mM $H_2O_2$ was divided by that of the control cell culture. This relative frequency value is shown in *Figure 1—figure supplement 1C*.

## Microfabrication

We fabricated microchamber arrays on glass coverslips by wet etching. The dimensions of each microchamber were 62 µm (w) × 47 µm (h) × 0.8 µm (d) for the array used in the experiments with a 100× objective (The number of microchambers in the array was 1240), and 53 µm (w) × 33 µm (h) × 0.8 µm (d) for the array used in the experiments with a 40× objective (The number of microchambers in the array was 22,302).

We first fabricated the photomask for the microchamber array with mask blanks (CBL4006Du-AZP, CLEAN SURFACE TECHNOLOGY) using a laser drawing system (DDB-201-TW, Neoark or DLS-50, Nano System Solutions). The photoresist on mask blanks was developed in NMD-3 (Tokyo Ohka Kogyo). The uncovered chromium (Cr)-layer was removed in MPM-E350 (DNP Fine Chemicals), and the remaining photoresist was removed by acetone. Lastly, the slide was rinsed in MilliQ water and air-dried.

For creating microchambers based on the pattern on the photomask, we first coated a 1,000-angstrom Cr-layer on a clean coverslip (NEO Micro glass, No. 1., 24 mm × 60 mm, Matsunami) by evaporative deposition and AZP1350 (AZ Electronic Materials) by spin-coating on the Cr-layer. After the pre-bake (95°C, 90 sec), we exposed the photoresist layer on the coverslip to UV light through the photomask using a mask aligner (MA-20, Mikasa). We developed the photoresist in NMD-3 and removed the part of Cr-layer uncovered by the photoresist in MPM-E350. The coverslip was soaked in buffered hydrofluoric acid solution (110-BHF, Morita Kagaku Kogyo) for 14 min 20 sec at 23°C for glass etching. The etching reaction was stopped by soaking the coverslip in milliQ water. The remaining photoresist and the Cr-layer were removed by acetone and MPM-E350, respectively.

We used a polydimethylsiloxane (PDMS) pad with bubble trap groove to control and change the culture conditions around the cells in the microchamber array. We followed the same procedure as that reported in *Yamauchi et al., 2022* to fabricate the PDMS pad.

## Chemical decoration of coverslip and cellulose membrane

To enclose the *E. coli* cells in the microchambers, we employed a method of attaching a cellulose membrane to a glass coverslip via biotin-streptavidin binding (*Inoue et al., 2001*). We chemically decorated glass coverslips with biotin and cellulose membranes with streptavidin. We followed the same protocol reported in *Yamauchi et al., 2022* to prepare biotin-decorated coverslips and streptavidin-decorated cellulose membranes.

## Preparation of conditioned media

As described below ('Sample preparation for time-lapse measurements'), we used the conditioned media to maintain the growth-suppressive conditions while preparing single-cell measurements of the cells sampled from the early and late stationary phases. We prepared 50 mL of the *E. coli* cell suspension at $OD_{600} = 0.005$ in the LB medium or at $OD_{600} = 0.01$ in the M9 medium following the same procedure of growth curve measurements and incubated the cell cultures at 37°C with shaking. At the pre-determined time (8 h for the early stationary phase condition and 24 h for the late stationary phase condition), we spun down the cells at 2,200×*g* for 12 min. The supernatant was again spun down at 2,200×*g* for 12 min, and its supernatant was filtered with a 0.2-μm syringe filter. We used this filtered medium as the conditioned medium.

To prepare the agarose blocks of conditioned media, we dissolved 2% (w/v) low-melt agarose in the conditioned media with a microwave and solidified it in a plastic dish. To use this for attaching a cellulose membrane to a coverslip, we excised a piece of the conditioned medium pad with a sterilized blade and placed it on the cellulose membrane.

## Sample preparation for time-lapse measurements

An overview of the sample preparation for the killing curve assay is shown schematically in *Figure 1C*. To prepare *E. coli* cells for the time-lapse single-cell observation, 5 μL of the glycerol stock of the *E. coli* cells was inoculated into 2 mL LB or M9 medium and incubated at 37 °C with shaking for 15 h (LB) or 16 h (M9) (first pre-culture). The cell culture was diluted to $OD_{600} = 0.005$ (in the case of LB medium), $OD_{600} = 5×10^{-6}$ (in the case of the 1000-fold more diluted condition in LB medium, *Figure 3—figure supplement 1*) or $OD_{600} = 0.01$ (in the case of M9 medium) in a 50 mL fresh medium and again incubated at 37°C with shaking. We sampled cells from the cell culture at the pre-determined time points (2~2.5 h for the post-exponential phase condition in LB; 6 h for the post-exponential phase condition grown from the diluted cell suspension; 8 h for the post-early stationary phase condition in LB; 24 h for the post-late stationary phase condition in LB; 4 h for the post-exponential phase condition in M9; and 24 h for the post-late stationary phase condition in M9; second pre-culture).

For the post-exponential phase conditions (both LB and M9), 1 μL of cell suspension in the second pre-culture was sampled from the culture and placed on the microchamber array. We put a streptavidin-decorated cellulose membrane on the cell suspension and removed an excess suspension with a clean filter paper. A small piece of an LB or M9 agarose block (2% (w/v) agarose) was placed on the cellulose membrane and incubated at 37°C for 15 min to allow the membrane to adhere to the coverslip surface tightly. After removing the agarose block, a PDMS pad was attached to the coverslip via a Frame-Seal Incubation Chamber (9 mm×9 mm, 25 μL internal volume; or 15 mm×15 mm, 65 μL internal volume, Bio-Rad). We connected the medium inlet and outlet of the PDMS pad to silicone tubes and immediately started to flow LB or M9 medium at a rate of 32 mL/h until the Frame-Seal Incubation Chambers and tubes were filled with the medium, and then at 2 mL/h using syringe pumps (NE-1000, New Era Pump Systems). After registering the microchamber positions, time-lapse measurements were started and continued as described in the following section.

For the early and late stationary phase samples, we followed the same protocol except that we used the conditioned medium agarose pads made from the cell cultures of the corresponding growth phases to attach the cellulose membrane. We also flowed the liquid conditioned medium while registering the microchamber positions to maintain the growth-suppressive conditions. We switched the flowing media from the conditioned media to the fresh medium at the start of the time-lapse measurements to arrange the same duration of regrowth before the Amp exposure across the replicate experiments.

## Time-lapse observation with the MCMA microfluidic device

We used Nikon Ti-E microscopes equipped with ORCA-flash camera (Hamamatsu Photonics) for the time-lapse observations. For the time-lapse observations with a 40× objective (Plan Apo $\lambda$, 40×/NA0.95), we acquired only bright-field images to monitor many microchamber positions simultaneously. For the time-lapse observations with a 100× objective (Plan Apo $\lambda$, 100×/NA1.45, oil immersion), we acquired bright-field for MG1655 and MF1, RpoS-mCherry fluorescence, and GFP fluorescence images for MF1. We used an LED illuminator (MCWHL2, Thorlabs) for fluorescence

excitation. We maintained the temperature around the microscope stage at 37°C with a thermostat chamber (TIZHB, Tokai Hit).

The microscope was controlled from a PC using Micromanager (*Edelstein et al., 2014*). The time-lapse intervals were 3 min in the measurements with the LB medium and 6 min in the measurements with the M9 medium.

In the time-lapse measurements of the cells in the LB medium, we first allowed the cells to grow in the microchambers flowing fresh medium at a rate of 2 mL/h for either 1.5 h (the cells from the exponential phase) or 2 h (the cells from the early and late stationary phases). We then changed the flowing media to the one containing 200 μg/mL of Amp rapidly at an increased flow rate of 32 mL/h for 3.75 min and then observed the cells flowing drug-containing medium at a rate of 2 mL/h for 6 h for MG1655 and 3.5 h for MF1. We quickly changed the flowing media back to the fresh medium without drug at an increased flow rate of 32 mL/h for 3.75 min and allowed the viable cells to re-pro-liferate by flowing the drug-free fresh medium at a rate of 2 mL/h. We typically observed cells for 12 h after antibiotic exposure. If cells begin to grow rapidly after treatment and their progeny grow into adjacent chambers, we stopped observing these chambers earlier than 12 h.

In the time-lapse measurements of the cells in the M9 medium, we allowed the cells to grow in the microchambers for 4 h flowing a fresh M9 medium. We then switched the flowing media to the one containing 200 μg/mL Amp or 1 μg/mL CPFX and observed the cells under the drug-exposed conditions for 6 h. We again switched the flowing media back to the M9 medium containing no drug and allowed the viable cells to regrow. We typically observed cells for 24 h after antibiotic exposure. As in the LB condition, if cells begin to grow rapidly after treatment and their progeny grow into adjacent chambers, we stopped observing these chambers earlier than 24 h.

## Elimination of long lagging cells in the post-exponential phase cell populations of MF1 in single-cell measurements

Based on the protocol described in the previous two sections, the upper bound for the fraction of long lagging cells in the post-exponential phase cell populations at the time of Amp exposure in the single-cell measurements can be estimated. In this protocol, cell populations enter stationary phase in the first pre-culture (*Figure 1C*). This stationary phase cell culture is diluted in fresh medium to a cell density corresponding to $OD_{600} = 0.005$ in the case of LB condition. For the experiment with MF1, we typically incubated this cell culture for 2.5 h at 37°C with shaking for the post-exponential phase condition. We maintained growth conditions while loading cells into the MCMA device by placing LB agarose block at 37°C and while registering microchamber positions for time-lapse measurements by flowing fresh LB medium into the device at 37°C. This preparation step before the time-lapse measurements took approximately 1.5 h. The pre-exposure period in the time-lapse measurements (*Figure 1D*) was 1.5 h. In total, cells were placed under exponential growth conditions for approximately 5.5 h prior to Amp exposure. Consequently, the lagging cell fraction present at the beginning of the second pre-culture should have reduced its relative frequency in the cell population to $2^{-16} \approx 10^{-5}$ assuming a mean doubling time of 20 min. This sets the upper bound for the fraction of lagging cells. In addition, not all lagging cells can produce persister cells. Therefore, the probability of finding non-growing persister cells coming from the lagging cell fraction in our experimental setup is low.

To further eliminate such long lagging cell fractions, we also performed experiments in which the second pre-culture was started from a 1000-fold diluted inoculum and the duration of the second pre-culture was extended by 3.5 h to allow the cell population to reach the same cell density in the exponential phase as in the standard post-exponential phase cell populations described above. This change in the pre-culture procedure reduces the frequency of long lagging cells by a factor of 1000 at the time of Amp treatment, and the probability of detecting non-growing persisters from long-lagging cells becomes virtually zero in our experimental setup.

## Image analysis

We used a custom Fiji (*Schindelin et al., 2012*) macro to analyze the time-lapse images acquired with the 40× objective or the 100× objective. Cell outlines were manually detected and represented as polygons. The macro assigns the cell lineages based on the segmented cell image sequences. We corrected the errors in the segmentation and lineage assignments manually. The cell size (area), circularity, and the connections in the cell lineages were exported into a text file for the data analysis.

We opened the images as a stack to find the persister cell lineages in acquired the time-lapse images. From the end time point, we played back the stacked images to detect the microchambers with the surviving cells that stably re-proliferated after the drug removal. We tracked back the persistent cell lineages and determined the existence of growth and cell divisions before the drug exposure.

To estimate the number of drug-exposed cells, we randomly selected 20 microchambers and counted the number of cells in each microchamber. We multiplied the averaged number of the cells in a microchamber by the number of the microchambers tracked in the time-lapse measurement. We used this number as the estimate of the total number of the drug-exposed cells in each experiment (*Figure 3D*, *Figure 3—figure supplement 1*).

## Data analysis

### Estimation of persister cell frequencies

We estimated the persister cell frequencies in *Figure 7B and C* from the population killing curves as the number of surviving cells at the time point of 6 h for MG1655 and 4 h for MF1 under the drug exposure relative to the initial number of cells at the onset of drug exposure. The error ranges of the persister cell frequencies were estimated by the error propagation of the standard errors of the numbers of cells at the two time points.

### Estimation of non-growing cell fractions in cellular populations

We analyzed the time-lapse images observed with the 100× objective. To determine the non-growing cell fraction, we focused on the time window of 90 min immediately before the Amp exposure. Cells without elongation and division were considered non-growing cells for MG1655. In addition to these criteria, we checked for the presence of GFP and RpoS-mCherry fluorescence signals for MF1.

### Quantification of pre-exposure growth rate of cell lineages in exponential phase

The pre-exposure growth rates of the cell lineages sampled from the exponential phase cultures were quantified by the mean division rates. For each single-cell lineage $\sigma$, the number of cell divisions before Amp exposure $D(\sigma)$ and the duration of pre-exposure period $T(\sigma)$ were obtained from the time-lapse images acquired with a 40× objective. The division rate of each cell lineage was estimated as $D(\sigma)/T(\sigma)$. The means and standard errors of division rate for the set of growing persistenter cells and for the set of non-persister cells for each condition were reported in the main text. We extracted all the pre-exposure growing persisters that could be analyzed, while we extracted independent non-persisters originating from different individual cells at the start of the time-lapse observation by selecting either of sister cells randomly at each cell division up to the onset of drug exposure.

### Fitting multiple exponential decay curves to the killing curves

We assumed that the population killing curves follow $f_n(t) = \sum_{j=1}^{n} a_j e^{-d_j t}$, where $n$ is the number of phases, $a_j$ is a positive constant, $d_j$ is a decay rate, and $t$ is a time spent under antibiotic exposure. We conducted non-linear least square fitting of the logarithms of the experimental data with $\ln f_n(t)$ using the Levenberg-Marquardt algorithm on Gnuplot. To determine the number of phases $n$, we calculated Aakaike's Information Criterion (AIC) as

$$\text{AIC} = N \ln(2\pi) + N + N \ln\left(\frac{\text{SSR}}{N}\right) + 2k, \qquad (2)$$

where $N$ is the number of data points, SSR is the sum of squares of residuals, and $k$ is the number of parameters. SSR was taken from the result of the fitting on Gnuplot. We adopted $n$ that gave the minimum AIC within the range of the initial value set of the parameters and with the constraints of $a_j > 0$ and $d_j > 0$.

## Viability check by PI staining

We found that many non-growing cells in the post-late stationary phase cell populations remained intact during the Amp exposure but did not regrow after the drug removal. We examined the viability

of the non-growing cells flowing the media containing 1 µg/mL PI in the time-lapse measurements. In these experiments, we used the MG1655 pUA66-P$_{rpsL}$-$gfp$ strain. The preparation of the cells for the time-lapse measurements was the same as that used for the cells sampled from the late stationary phase, except that we flowed the media containing PI from the time point of adding Amp.

## Materials availability

*E. coli* cell strains and vectors used in this study are available from the authors upon request.

## Acknowledgements

We acknowledge Nathalie Balaban for comments on the initial version of the manuscript; Shoji Takeuchi for his support of the microfabrication facility; and members of the Wakamoto lab for discussion. This work was supported by JST CREST Grant Number JPMJCR1927 (YW); JST ERATO Grant Number JPMJER1902 (YW); NIH Grant Number R01-GM097356 (EK); and Japan Society for the Promotion of Science KAKENHI Grant Number 17H06389, 19H03216, and 24H00552 (YW).

## Additional information

### Funding

| Funder | Grant reference number | Author |
|---|---|---|
| Japan Science and Technology Agency | 10.52926/jpmjcr1927 | Yuichi Wakamoto |
| Japan Science and Technology Agency | 10.52926/jpmjer1902 | Yuichi Wakamoto |
| National Institute of General Medical Sciences | R01-GM097356 | Edo Kussell |
| Japan Society for the Promotion of Science | 17H06389 | Yuichi Wakamoto |
| Japan Society for the Promotion of Science | 19H03216 | Yuichi Wakamoto |
| Japan Society for the Promotion of Science | 24H00552 | Yuichi Wakamoto |

The funders had no role in study design, data collection and interpretation, or the decision to submit the work for publication.

### Author contributions

Miki Umetani, Conceptualization, Formal analysis, Validation, Investigation, Methodology, Writing – original draft, Writing – review and editing; Miho Fujisawa, Conceptualization, Formal analysis, Investigation, Methodology; Reiko Okura, Formal analysis, Investigation, Methodology; Takashi Nozoe, Hidenori Nakaoka, Investigation, Writing – review and editing; Shoichi Suenaga, Investigation; Edo Kussell, Supervision, Funding acquisition, Writing – review and editing; Yuichi Wakamoto, Conceptualization, Formal analysis, Supervision, Funding acquisition, Investigation, Methodology, Writing – original draft, Writing – review and editing

### Author ORCIDs

Miki Umetani 
Takashi Nozoe 
Shoichi Suenaga 
Hidenori Nakaoka 
Edo Kussell 
Yuichi Wakamoto 

### Decision letter and Author response

Decision letter https://doi.org/10.7554/eLife.79517.sa1

Author response https://doi.org/10.7554/eLife.79517.sa2

## Additional files

### Supplementary files
MDAR checklist

### Data availability
All the data and data analysis codes for generating figures have been deposited to Dryad.

The following dataset was generated:

| Author(s) | Year | Dataset title | Dataset URL | Database and Identifier |
|---|---|---|---|---|
| Umetani M, Fujisawa M, Okura R, Nozoe T, Suenaga S, Nakaoka H, Kussell E, Wakamoto Y | 2025 | Data from: Observation of persister cell histories reveals diverse modes of survival in antibiotic persistence | https://doi.org/10.5061/dryad.s1rn8pkb1 | Dryad Digital Repository, 10.5061/dryad.s1rn8pkb1 |

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
