## [Editor Report]

The manuscript reports on the single cell evaluation of *E. coli* persisters under antibiotics. Typically, persisters are rare cells and only very few have been directly observed in non mutated strains. Therefore, the current work adds an important contribution by mapping a higher numbers of persisters and by convincingly identifying different persister phenotypes. The main conclusions are along previous reports, namely that persisters under β-lactams are mainly non-growing cells from stationary conditions, but if cultures are carefully kept away from stationary phase conditions, the few residual persisters observed in this study display different phenotypes involving growing cells and L-forms.

---

## [Decision Letter]

**Decision letter after peer review:**

Thank you for submitting your article "Observation of non-dormant persister cells reveals diverse modes of survival in antibiotic persistence" for consideration by *eLife*. Your article has been reviewed by 3 peer reviewers, one of whom is a member of our Board of Reviewing Editors, and the evaluation has been overseen by Naama Barkai as the Senior Editor. The following individual involved in the review of your submission has agreed to reveal their identity: Thomas Julou (Reviewer #3).

Essential Revisions (for the authors):

The reviewers have done a thorough work in pointing out the ways to address major weaknesses, and these should be seriously addressed. In particular:

1) As the reviewers point out, it seems that important data has not been analyzed, which would shed new light on the persister's physiology: whether the non-dormant persister cells do grow during the antibiotic treatment or arrest their growth, whether there is a higher probability within a lineage to persist, etc. More generally, a more quantitative analysis of the different modes of survival is needed in order to represent more than an incremental contribution to previous observations of growing persisters.

2) The data on the problematic RpoS reporter should be either taken out or validated: is RpoS functional and how is the reporter itself affecting the growth and survival?

3) The methods should be more detailed, especially the history of the culture before the persister assays. Emphasis should be put on whether the dormant cells may be from lagging bacteria, as a few hours are not enough to get rid of all lagging bacteria.

4) Specific comments of the reviewers regarding to reference to literature that considers also non-dormant persisters.

*Reviewer #1 (Recommendations for the authors):*

– For example, there is no data on whether the non-dormant persister cells do grow during the antibiotic treatment or arrest their growth. Additional quantifications on the number of non-dormant persister that make L-forms, arrest their growth upon antibiotic exposure, or continue to grow may add some insight into the different mechanisms of survival.

– One main comment is that the authors claim at several instances in the text that the community attributes persistence mainly to dormant cells and their observations contradict this view. First, it has been recognized early on that not all persister cells are dormant; second, by following the type of persister bacteria that they observe along the growth curve, the authors show, in agreement with previous results, that very "clean" conditions are needed to obtain more not-dormant persisters than dormant ones. This means that the authors actually support the common view, which is that non-dormant persisters can be found, but in typical conditions, dormant ones will be more common. Therefore, it is advised to re-phrase those claims.

– The authors mention "persister lineages" while the observation is of only one bacterium each time surviving the antibiotic treatment. They do not show any propensity for survival within a lineage so the term may be misinterpreted. It would therefore be clearer to simply call them "persister cells". If there are instances where sisters or cousin cells survive in unison, indicating "persister lineages", it may be worthwhile to show them.

*Reviewer #2 (Recommendations for the authors):*

The manuscript by Umetani et al. is interesting and provides additional support to early observations showing that persister cells can arise from actively growing cells. The authors tested different conditions and analyzed a substantial number of persister cells, which is not an easy task.

I have some questions/comments regarding the RpoS-mCherry fusion:

The fusion is inserted at the rpos chromosomal locus, replacing the wild-type rpoS allele as far as I understand. The question is whether the fusion is functional or disrupts the activity of RpoS and makes the cells RpoS-. The authors should test whether RpoS is still functional. An easy assay is to measure the survival of the strain in the presence of 20 mM H2O2 in a minimal M9 medium with glucose. An RpoS- strain shows a decrease in survival in these conditions. The authors might also repeat some experiments with an RpoS defective strain or a RpoS+ strain (depending on the genotype of the strain used) to test whether RpoS does influence persistence in their experimental settings.

To validate the RpoS fusion, the authors should also provide some data regarding the increase of RpoS levels as a function of growth by simply measuring fluorescence at the population level. The expectation is that fluorescence will increase as a function of time.

I might have missed the details of the methods used by the authors to eliminate the carryover of the long-lagging dormant cells in exponential growing populations. This should be added or better explained. In this experiment, the authors observed spontaneous dormant cells without any apparent trigger. What is the author's hypothesis? Are these cells dead, were they growing at some point during the experiment and stopped growing, or do they show some detectable levels of RpoS? What would be the mechanism(s)?

The section 'non-dormant persisters are dominant even in minimal medium'. I'm not sure why the authors added 'even'. What were their expectations?

In the experiment in M9 minimal medium, the authors used 200 ug/mL of ampicillin and treated it for 6h. What is the reason? In Figure S4 the authors also used 200 ug/mL of ampicillin while in the experiments presented before 100 ug/mL was used as far as I understand.

In the section on the stationary phase, the authors found many dormant cells that did not PI stained. The authors conclude that they are probably viable and did not regrow during the time cells were observed under the microscope. The question here is whether those cells could be dead even if they do not PI stained? Do these cells contain high RpoS levels?

Ref 23 cited in this section is a BioRxiv manuscript, I don't know if it can be cited.

The next section starts with the measurement of dormant cells in the different conditions tested and provides an equation to predict the frequency of surviving cells after treatment. Although I'm not competent to judge the equation, the conclusion reached by the authors is that the longer in the stationary phase the better sit is for the survival of ampicillin. Some papers from the Tenson group came to this conclusion and should be cited e.g. Luidalepp et al. J. Bact 2011; Joers and Tenson, Sci Rep 2016; Joers et al. J Bact 2010. It would certainly be interesting to discuss these data in light of the present single-cell analysis.

Dormant cells from the late stationary phase express high levels of RpoS that seem to form aggregates. Although it's plausible, it would be nice to show some evidence. The authors could try the IbpA fusion that colocalizes with aggregates – recently published by the group of Aertsen (Govers et al., PLoS Biol 2018). The assumption here is that in the early or late stationary phase, the cells with the highest RpoS levels are not the persisters and that RpoS is aggregated in those cells. What is the idea? Do aggregates somehow prevent cells to regrow after the ampicillin treatment? Are these cells dead? Are the RpoS-specific aggregates somehow toxic? Or is it a general phenomenon evidenced here by the RpoS fluorescent fusion? In other words, if the authors use another fluorescent marker reporting a protein highly expressed in the stationary phase, will they observe the same phenomenon.

In any case, it would be interesting to characterize these cells further and link the data with those presented in other papers, e.g. several papers from the Michiels group that link aggregates and persistence (Dewachter et al., mBio 2021; Bollen et al., Front Mol Biosci 2021).

Figure 4F is not clear, the red dots are barely visible.

The last section deals with persistence to ciprofloxacin and parallels the data observed by Goormaghtigh and Van Melderen Ref 20. In the reference list, the last name of the last author should be Van Melderen, not Melderen.

*Reviewer #3 (Recommendations for the authors):*

I couldn't find the information on how long cells are exposed to fresh media after antibiotic treatment. It is important for the reader that this phase is long enough, so it should be mentioned in relevant places.

Several semantic issues:

The word dormant is used here for "which has not restarted growing during the 2 h in LB before the treatment". This type of growth arrest where cells are already exposed to fresh media but not yet regrowing (hence more likely to have resumed some metabolic activity) is almost universally referred to as "lag" and this would be a much more appropriate wording. Dormancy is a more loosely defined term that seems to imply for most people a prolonged growth arrest.

The authors often refer to persister lineages – in general one rather likes to think of persister cells which each give rise to a lineage (where most cells are expected to be sensitive).

The article is mostly about persistence to ampicillin which does not represent all antibiotics, hence the title and writing would benefit from being toned down.

The rationale for fitting the killing curve in Figure 1A with a triple exponential is unclear.

By comparing Figure 3B and 3C, it is remarkable that the shortest lags in Late Stationary are shorter than the shortest lags in Early Stationary which is very counter-intuitive and likely to point to an experimental issue. In addition, the mode of the elongation rate for growing cells in Figure 3E is lower than in Figure 3D and 3F which is also unexpected.

In contrast to most manuscripts I review, this manuscript was exceptionally well proofread and features virtually no typos. I only spotted " the same protocol as for growth curve measurements" (p. 17), as well as the caption of Supp Video 11.

Coming back to the first major concern expressed in the Public Review, I explain hereafter why I believe that "dormant" cells are expected to possibly be in very different states in different conditions (i.e. they do not have the same state in terms of gene expression, nor of metabolic state). I also comment on what is expected following Balaban et al. 2019; note that I assume for this that the state leading to spontaneous persistence also happens after the stationary phase and that persisters in the stationary phase can hence be decomposed into spontaneous and triggered. Importantly this has not been established and another description might well be more relevant. The 4 main conditions studied here are:

Exponential (diluted):

Preculture: 6 h growth in LB after ~5e5x dilution (hence 15-16 divisions) from overnight culture (5 to 10 h in stationary phase).

Assay: 1.5 h growth in LB followed by 3.5 h treatment. This condition is designed specifically to get rid of non-growing cells which might have been carried over from the last stationary phase; hence only spontaneous persistence is expected.

Exponential:

Preculture: 2.5 h growth in LB after ~500x dilution (hence 6-7 divisions) from overnight culture (5 to 10 h in stationary phase).

Assay: same (1.5 h growth in LB followed by 3.5 h treatment). Given the relatively low dilution factor from the overnight culture, one expects a mixture of spontaneous persistence and possibly some triggered persistence carried over from the last stationary phase.

Early Stationary:

Preculture: 8 h growth in LB after ~500x dilution (hence 4 h in stationary phase) from overnight culture (5 to 10 h in stationary phase).

Assay: 2 h growth in LB followed by 3.5 h treatment. Since most cells restart growing during the 2 h in LB at the start of the assay (Figure 3E), one does not expect tolerance by lag to play a role; however since cells divide only 1 to 3 times since the last stationary phase, triggered persistence is expected to occur at a higher frequency than spontaneous persistence.

Late Stationary:

Preculture: 24 h growth in LB after ~500x dilution (hence 20 h in stationary phase) from overnight culture (5 to 10 h in stationary phase).

Assay: same (2 h growth in LB followed by 3.5 h treatment). For a quarter of the population that has not exited the lag during the 2 h in LB at the start of the assay (Figure 3F), tolerance by lag is expected to be the main mode of survival. For the other three quarters, triggered persistence is expected to happen at a higher frequency than spontaneous persistence (note that the rate is probably different from Early Stationary).

Overall non-growing cells seen in Exponential (diluted) are expected to be only due to "bad luck" (Levin, Concepción-Acevedo, and Udekwu 2014). For the Exponential condition, non-growing cells are expected at a similar frequency due to "bad luck", as well as possibly due to carry over from the previous stationary phase (very long lag); note that such carryover doesn't seem to change the frequency of growing and non-growing persisters (Figure 1 of this review).

In the Early Stationary condition, most cells have restarted growing only recently hence some proteins/metabolites involved in persistence are expected at a higher level, which will lead to triggered persistence. Finally growing cells in the Late Stationary condition have spent longer in the stationary phase, which means that other proteins/metabolites might be present and/or at higher levels; this manifest in their longer lags before restating growing in LB (visible from the higher fraction of non-growing cells after 2 h).

Regarding Late Stationary, I anticipate that the authors may claim that it cannot correspond to tolerance since the killing curve (yellow in Figure 2B) doesn't display a decrease in the initial death rate which is the hallmark of tolerance. I indeed don't know how to reconcile this observation with Figure 3F which shows that 26% of the population is still in the lag phase after the stationary phase; however since this high fraction stands as the main difference between Early Stationary (Figure 3E) and since non-growing bacteria are notoriously less sensitive to β-lactams such as ampicillin, one can only be tempted to hypothesize that these cells are indeed tolerant. In fact, I would like to suggest that maybe the killing curve would look completely different for Late Stationary (but not for the other 2 conditions) if prolonged far beyond the lag time of this strain out of the stationary phase.

Balaban, N.Q. et al. (2019) 'Definitions and guidelines for research on antibiotic persistence', Nature Reviews. Microbiology, 17, pp. 441-448. doi:10.1038/s41579-019-0196-3.

Goormaghtigh, F. and van Melderen, L. (2019) 'Single-cell imaging and characterization of *Escherichia coli* persister cells to ofloxacin in exponential cultures', Science Advances, 5(6), p. eaav9462. doi:10.1126/sciadv.aav9462.

Levin, B.R., Concepción-Acevedo, J. and Udekwu, K.I. (2014) 'Persistence: a copacetic and parsimonious hypothesis for the existence of non-inherited resistance to antibiotics', Current Opinion in Microbiology, 21, pp. 18-21. doi:10.1016/j.mib.2014.06.016.

Maisonneuve, E., Castro-Camargo, M. and Gerdes, K. (2013) '(p)ppGpp controls bacterial persistence by stochastic induction of toxin-antitoxin activity', Cell, 154(5), pp. 1140-1150. doi:10.1016/j.cell.2013.07.048.

Patange, O. et al. (2018) '*Escherichia coli* can survive stress by noisy growth modulation', Nature Communications, 9(1), p. 5333. doi:10.1038/s41467-018-07702-z.

---

## [Author Response]

Essential Revisions (for the authors):The reviewers have done a thorough work in pointing out the ways to address major weaknesses, and these should be seriously addressed. In particular:1) As the reviewers point out, it seems that important data has not been analyzed, which would shed new light on the persister's physiology: whether the non-dormant persister cells do grow during the antibiotic treatment or arrest their growth, whether there is a higher probability within a lineage to persist, etc. More generally, a more quantitative analysis of the different modes of survival is needed in order to represent more than an incremental contribution to previous observations of growing persisters.

We thank the reviewers for the important suggestion.

To address this issue, we have included more detailed analyses of single-cell dynamics of growth and cell shape over the course of pre-exposure, exposure, and post-exposure periods (Figure 2D and H, Figure 4B and D, Figure 4 —figure supplement 1 and 2, Figure 5B and D, Figure 5 —figure supplement 1, Figure 8B and D, and Figure 8 —figure supplement 1). As a result, we show not only the presence of persister cells that were growing normally prior to the antibiotic treatment, but also their heterogeneous growth response to the treatment. For example, in the case of ampicillin (Amp) treatment, some cells continue to grow and undergo multiple rounds of cell body fission with L-form-like morphologies and return to normal rod shapes after drug removal (Figure 4D); some other cells suppressed their growth significantly typically after the first divisions during the Amp treatment (Figure 4B); and the other cells exhibited the growth behaviors between these two, with intermittent growth and growth arrest (Figure 4 —figure supplement 1). Furthermore, the transition to L-form-like morphologies also occurred during the process of regaining normal growth in the postexposure period. Thus, the new analysis reveals the heterogeneous growth and cell shape dynamics induced by Amp treatment among the persisters growing before the Amp treatment.

In the case of ciprofloxacin (CPFX) treatment, many persister cells became filamentous after the treatment (Figure 8A and B), as previously reported for a similar fluoroquinolone antibiotic, ofloxacin (ref. 20). However, we also identified surviving cells that resumed normal division after the treatment without becoming filamentous (Figure 8C and D). Although it is a correlated phenotype, post-treatment filamentation is not always required for persister cells to regain normal growth.

We also quantified and compared the pre-treatment division rates between the persister cells growing prior to antibiotic treatments and the non-surviving cells. Although the persister cells of MG1655 against Amp treatment tended to be slow growers in the post-exponential phase cell populations in M9 medium compared to non-survivors (Figure 5E), such a trend was not observed for the other conditions under which we detected growing persister cells (post-exponential phase in LB medium against Amp and post-exponential phase in M9 medium against CPFX) (Figure 4E and 8E). Therefore, growing persister cells of MG1655 found in these treatment conditions appear to circumvent the fitness trade-off in the antibiotic-free environments.

We believe that the additional detailed information on single-cell dynamics obtained for different strains, growth phases, and antibiotics presented in the revised manuscript would be valuable to the field and advance our understanding of antibiotic persistence. For further details, please see our point-by-point replies to the specific comments by the reviewers.

2) The data on the problematic RpoS reporter should be either taken out or validated: is RpoS functional and how is the reporter itself affecting the growth and survival?

We thank the reviewers for pointing out the problem with the RpoS reporter. As suggested by reviewer 2, we have tested the functionality of RpoS by evaluating its sensitivity to H2O2 oxidative stress. We found that the reporter strain MF1 is significantly more sensitive to oxidative stress than the wildtype (MG1655) and as sensitive as the ∆*rpoS* strain (Figure 1 —figure supplement 1C).

Although the induction of RpoS in the MF1 strain followed similar kinetics reported previously (Figure 1 —figure supplement 1A and B, ref. 32), the result of the oxidative stress test suggests that the RpoS functions are impaired.

After confirming the problem with the reporter strain, we repeated almost all the experiments with the MG1655 strain. As shown, the main results and conclusions are unchanged, such as the dominance of persisters growing normally before antibiotic treatment in the post-exponential phase cell populations (Figure 3 and Figure 3 —figure supplement 1), L-form-like morphological changes by Amp treatment in persisters growing normally before treatment (Figure 4C and 5C), and changes in the dominant survival modes against Amp treatment depending on the sampling growth phases (Figure 3 and Figure 3 —figure supplement 1).

Although the functional problem of the RpoS reporter was confirmed, we decided to keep and present most of the results of the MF1 strain, except those on the correlations between RpoS expression level and survival, because the results are informative and indicate that intact RpoS function is not a prerequisite for the occurrence of antibiotic persistence against Amp and CPFX, but affects the killing kinetics of the cell population and the frequency of persister cells in the populations (Figure 1 —figure supplement 3 and Figure 6 —figure supplement 1). Indeed, when the population killing curves of the post-exponential and post-early stationary phase cultures are compared between the MG1655 and MF1 strains, the number of surviving cells of MG1655 against Amp treatment is initially larger than that of MF1, but becomes smaller with prolonged exposure (> 3 h) (Figure 1 —figure supplement 3A and Figure 6 —figure supplement 1B). On the other hand, the number of surviving cells of MG1655 is consistently larger than that of MF1 for the postlate stationary phase cell populations (Figure 6 —figure supplement 1B). These results demonstrate both positive and negative contributions of RpoS regulation to antibiotic persistence. We believe this information is of interest to the field and have decided to include the results from the MF1 strain in the revised manuscript.

3) The methods should be more detailed, especially the history of the culture before the persister assays. Emphasis should be put on whether the dormant cells may be from lagging bacteria, as a few hours are not enough to get rid of all lagging bacteria.

More detailed methods have been added to the main text. In particular, to clarify the history of the culture before the persister assay with the MCMA microfluidic device, an illustration of the sample preparation and experimental procedure that was shown as a supplementary figure in the original manuscript has been moved to a main figure (Figure 1C and D). In addition, we have added an illustration of the sample preparation for the killing curve assay in batch cultures (Figure 6A).

We thank the reviewer for pointing out that non-growing cells in the post-early stationary and post-late stationary phase cultures should be referred to as “lagging cells” rather than “dormant cells.” Since all cells were maintained in a non-growing state by the corresponding stationary phase conditioned medium until the start of the time-lapse measurement in the post-early stationary and post-late stationary phase conditions, non-growing cells found in these experiments are by definition lagging cells that failed to rapidly resume growth. We now explicitly state this in Results (L294-297).

In the original manuscript, we technically referred to the cells that did not grow and divide before the antibiotic treatments as “dormant cells,” but we agree that the term may be confusing because we did not distinguish truly metabolically inactive dormant cells from the other forms of non-growing cells. Therefore, we now avoid referring to these non-growing cells as “dormant cells” throughout this paper. Accordingly, we have changed the title of this paper from “Observation of non-dormant persister cells reveals diverse modes of survival in antibiotic persistence” to “Observation of persister cell histories reveals diverse modes of survival in antibiotic persistence”.

4) Specific comments of the reviewers regarding to reference to literature that considers also non-dormant persisters.

We have modified the Introduction to more appropriately place this research in the context of previous studies regarding the contribution of non-dormant cells to antibiotic persistence, based on the specific comments of reviewers 1 and 3.

Reviewer #1 (Recommendations for the authors):– For example, there is no data on whether the non-dormant persister cells do grow during the antibiotic treatment or arrest their growth. Additional quantifications on the number of non-dormant persister that make L-forms, arrest their growth upon antibiotic exposure, or continue to grow may add some insight into the different mechanisms of survival.

We appreciate the suggestion of an important analysis. To address this issue, we have analyzed the single-cell growth dynamics of both growing and non-growing persister cells in more detail and found heterogeneous dynamics of growing persisters during the exposure and post-exposure periods. In LB, some growing persisters against Amp in the post-exponential phase cell populations continued growth and cell body fission with L-form-like cell shapes, while some other growing persisters stopped growing after the first division under Amp exposure. The other growing persisters exhibited dynamics between these two, with intermittent growth and growth arrest (Figure 4A-D, Figure 4 —figure supplement 1, and Video 4). Furthermore, L-form-like cell body fission (division) was also observed in these growing persister cells during the processes of regaining normal rod shape and growth pattern. The responses of growing persisters to Amp treatment in M9 were also heterogeneous, with some cells continuing to grow and divide with Lform-like cell shape and the other cells suppressing their growth to varying degrees (Figure 5, Figure 5 —figure supplement 1, and Video 8). We found that many cells exhibited L-form-like cell body fission as they regained normal cell shape and division pattern. In addition, we found growing persisters that became filamentous after drug removal (Figure 5 —figure supplement 1 and Video 8), as observed in the growing persisters against CPFX. These results are now included in the revised manuscript.

Due to the difficulty in clearly distinguishing these different dynamics, we have decided that it is not possible to accurately quantify the frequencies of the different patterns of growing persister cell dynamics at this time. Instead, we show all single-cell dynamics of persister cells as figure supplements to provide information on the heterogeneity of the dynamics. Investigating whether these different persister cell dynamics are underpinned by different molecular mechanisms is an interesting and important question, but we have concluded that identifying these mechanisms is beyond the scope of the current study.

– One main comment is that the authors claim at several instances in the text that the community attributes persistence mainly to dormant cells and their observations contradict this view. First, it has been recognized early on that not all persister cells are dormant; second, by following the type of persister bacteria that they observe along the growth curve, the authors show, in agreement with previous results, that very "clean" conditions are needed to obtain more not-dormant persisters than dormant ones. This means that the authors actually support the common view, which is that non-dormant persisters can be found, but in typical conditions, dormant ones will be more common. Therefore, it is advised to re-phrase those claims.

We appreciate these important comments.

Regarding the first point, we have modified the Introduction and many other sections to place our study in a more appropriate context based on the previous studies demonstrating the contributions of growing persisters.

Regarding the second point, to address whether very “clean” conditions are needed to obtain more growing persisters than non-growing ones, we performed additional experiments to expose CPFX to post-late stationary phase cell populations in M9 and found that all persisters for which we could identify their pre-exposure growth trait were growing persisters. This result suggests that the dominance of growing persisters is not necessarily limited to clean conditions and may depend on the type of antibiotics and conditions. We now present this new result in Figure 3 and Video 18 and have also added a discussion to the main text (L490-502).

– The authors mention "persister lineages" while the observation is of only one bacterium each time surviving the antibiotic treatment. They do not show any propensity for survival within a lineage so the term may be misinterpreted. It would therefore be clearer to simply call them "persister cells". If there are instances where sisters or cousin cells survive in unison, indicating "persister lineages", it may be worthwhile to show them.

We have changed “persister lineages” to “persister cells” in the revised manuscript.

In our previous analysis with the MF1 strain in the original manuscript, we did not find any cases where sister or cousin cells survived simultaneously. However, in the newly added experiments with the MG1655 strain, we identified one such pair of persister cells to Amp in the postexponential phase cell populations in M9 medium (growing persister cell 1 in Figure 5B and growing persister cell 4 in Figure 5 —figure supplement 1). Given the extremely low frequency of persister cells, the chance of observing such a surviving pair is almost negligible if the survival fate is independent for all cells. Therefore, the fact that we observed such a pair in this observation suggests that the fate of persister cells may have a lineage correlation. However, our observation is limited to one example, and verification of this speculation requires single-cell analysis at much higher throughput. We have now mentioned this result and the limitation of the current analysis in the main text (L251-261).

Reviewer #2 (Recommendations for the authors):The manuscript by Umetani et al. is interesting and provides additional support to early observations showing that persister cells can arise from actively growing cells. The authors tested different conditions and analyzed a substantial number of persister cells, which is not an easy task.I have some questions/comments regarding the RpoS-mCherry fusion:The fusion is inserted at the rpos chromosomal locus, replacing the wild-type rpoS allele as far as I understand. The question is whether the fusion is functional or disrupts the activity of RpoS and makes the cells RpoS-. The authors should test whether RpoS is still functional. An easy assay is to measure the survival of the strain in the presence of 20 mM H2O2 in a minimal M9 medium with glucose. An RpoS- strain shows a decrease in survival in these conditions. The authors might also repeat some experiments with an RpoS defective strain or a RpoS+ strain (depending on the genotype of the strain used) to test whether RpoS does influence persistence in their experimental settings.

We thank the reviewer for bringing this critical issue to our attention and for suggesting an important experiment. We have performed the suggested experiment with the MF1 strain and tested its survival to the H2O2 oxidative stress. As mentioned in our reply to Essential Revision 2, we have found that the MF1 strain is as sensitive to the oxidative stress as the Δ*rpoS* strain. Therefore, the RpoS function seems to be affected by the tagged mCherry. We now show this important result in Figure 1 —figure supplement 1C.

Given this result, we decided to repeat almost all of the experiments with the wild-type strain MG1655 strain and confirmed that most of the conclusions are unchanged, such as the dominance of growing persisters to Amp treatment in the post-exponential phase cell populations. Notably, we did not detect any non-growing (dormant) persisters in the post-exponential phase cell populations of the MG1655 strain (Figure 3). Consistent with the observation with the MF1 strain, non-growing (dormant) persisters were dominant in the post-early stationary phase and post-late stationary phase cell populations of the MG1655 strain (Figure 3).

Again, consistent with the result with the MF1 strain, no non-growing persisters to CPFX treatment were detected in the post-exponential phase cell populations of the MG1655 strain (Figure 3). Interestingly, a newly added experiment revealed that the persisters to CPFX in the postlate stationary phase cell populations were all growing cells, in contrast to the results for Amp treatment (Figure 3). Therefore, these results demonstrate that growing persisters can be dominant even in post-late stationary phase cell populations, depending on the antibiotic type and culture conditions.

We removed the results of the correlation between RpoS expression levels and survival of individual cells of the MF1 strain because it may not reflect the relationship of a fully functional RpoS. Future studies with more appropriate RpoS reporters will quantitatively clarify the role of RpoS in antibiotic persistence.

To validate the RpoS fusion, the authors should also provide some data regarding the increase of RpoS levels as a function of growth by simply measuring fluorescence at the population level.The expectation is that fluorescence will increase as a function of time.

Again, we thank the reviewer for suggesting an important experiment. We measured the expression levels of the RpoS-mCherry fusion along the growth curves in LB and M9 media and confirmed that the expression levels follow similar kinetics as previously reported (ref. 32). We now present this result in Figure 1 —figure supplement 1A and B.

However, as mentioned above, the oxidative stress test confirmed the problem with the RpoSmCherry fusion protein (Figure 1 —figure supplement 1C). Therefore, we have removed the results that relied on RpoS-mCherry expression levels from the revised manuscript.

Despite the problem with the RpoS-mCherry fusion protein, the population killing curves of the MF1 strain still show antibiotic persistence (Figure 1 —figure supplement 3 and Figure 6 —figure supplement 1). Notably, we found that the number of surviving cells of the MF1 strain to long Amp treatment is greater than that of the MG1655 strain in the post-exponential and post-early stationary phase cell populations in LB (Figure 1 —figure supplement 3A and Figure 6 —figure supplement 1A), whereas it is consistently lower than that of the MG1655 strain in the post-late stationary phase cell populations (Figure 6 —figure supplement 1B). Therefore, RpoS has both beneficial and detrimental effects on antibiotic persistence depending on culture conditions. We think this information is important to the field and have decided to include these results in the revised manuscript.

I might have missed the details of the methods used by the authors to eliminate the carryover of the long-lagging dormant cells in exponential growing populations. This should be added or better explained. In this experiment, the authors observed spontaneous dormant cells without any apparent trigger. What is the author's hypothesis? Are these cells dead, were they growing at some point during the experiment and stopped growing, or do they show some detectable levels of RpoS? What would be the mechanism(s)?

We have added a more detailed explanation of the experiment to eliminate the carryover of long lagging non-growing cells in the post-exponential phase cell populations of the MF1 strain to the main text in both the Results (L161-178) and Methods (Page 38-39) sections. Additionally, we now show an illustration of the sample preparation procedure for the single-cell measurement in Figure 1C and D so that readers can understand the rationale behind the approach to eliminate the carryover of the long-lagging cells.

We first remark that no non-growing (dormant) persister cells were detected in the newly added experiments with the MG1655 strain (Figure 3). Therefore, the carryover issue is not relevant to the result with the MG1655 strain.

In the experiments with the MF1 strain, we detected two non-growing persisters in the post exponential phase cell populations (Figure 3 —figure supplement 1). The question here is whether these non-growing persisters were generated in the stationary phase of the first pre-culture from the glycerol stock, or whether they resumed growth in the second pre-culture but stopped growing before the start of the time-lapse experiment (see Figure 1C and D for the sample preparation procedure).

Our sample preparation procedure for the time-lapse microscopy with the MCMA device consisted of the following pre-culture steps:

1. First pre-culture: 5 µL of the glycerol stock was inoculated into 2 mL of fresh LB medium and incubated at 37°C with shaking for 15 hours. During this first pre-culture, the *E. coli* cell population entered stationary phase.

2. Second pre-culture: A cell suspension from the first preculture was reinoculated into LB medium with the initial cell density adjusted to OD600 = 0.005. This second pre-culture sample was incubated at 37°C with shaking for 2.5 hours. After 2.5 hours, cell populations are in exponential phase.

3. Loading of cells into microchambers and on-device pre-culture prior to Amp treatment: Cells in the exponential phase in the second pre-culture was taken and enclosed in the microchambers. Immediately after sealing the microchambers with a semipermeable cellulose membrane, we started to flow LB medium in the microfluidic device, keeping the temperature at 37°C. We kept the LB medium flowing in the device while registering the image acquisition positions, as well as after starting the time-lapse observation of individual cells prior to Amp treatment. Typically, approximately three hours elapsed from the time cells were sampled from the second pre-culture to the start of Amp treatment.

Following this procedure, since the cell populations were placed under growing conditions in LB medium for approximately 5.5 hours prior to Amp treatment, the lagging cell fraction present at the beginning of the second pre-culture should have reduced its relative frequency in the cell population to ~2^-16^ = ~10^-5^. In addition, not all lagging cells can produce persister cells. Therefore, the probability of finding non-growing persister cells coming from the lagging cell fraction in our experimental setup is low.

However, to further eliminate such long-lagging cell fraction, we modified the sample preparation step (2) and started the second pre-culture from a 1000-fold lower initial inoculum and extended the duration of the second pre-culture by ~3.5 hours so that the cell population reaches the same cell density in the exponential phase when cells are sampled in the step (3). This change in the pre-culture procedure reduces the frequency of long-lagging cells by a factor of 1000 at the time of Amp treatment, and the probability of detecting non-growing persisters from long-lagging cells becomes virtually zero in our experimental setup. Nevertheless, we still found an approximately equal fraction of non-growing persisters, suggesting that these persisters were not from the longlagging cell fraction and that they were generated during the second pre-culture, cell loading, or image acquisition position registration.

The section 'non-dormant persisters are dominant even in minimal medium'. I'm not sure why the authors added 'even'. What were their expectations?

Removed ‘even’ from the section title.

In the experiment in M9 minimal medium, the authors used 200 ug/mL of ampicillin and treated it for 6h. What is the reason? In Figure S4 the authors also used 200 ug/mL of ampicillin while in the experiments presented before 100 ug/mL was used as far as I understand.

We determined the duration of antibiotic treatment in the time-lapse experiments by referring to the population killing curves in batch cultures (Figure 1 —figure supplement 3C-E). We chose the antibiotic treatment durations at which *E. coli* cell populations were well into the persistent phase i.e., after the slope of the log population killing curve had shifted upward.

Although some previous studies have used the Amp concentration of 100 µg/mL in M9 medium (e.g., ref. 36), the same Amp concentration, 200 µg/mL, was used in both LB and M9 media in this study to allow comparison of results between different media. In both media, 200 µg/mL of Amp is well above the MICs of both MG1655 and MF1 strains (Figure 1 —figure supplement 2). We have clarified this point in the revised manuscript (L206-216).

In the section on the stationary phase, the authors found many dormant cells that did not PI stained. The authors conclude that they are probably viable and did not regrow during the time cells were observed under the microscope. The question here is whether those cells could be dead even if they do not PI stained? Do these cells contain high RpoS levels?Ref 23 cited in this section is a BioRxiv manuscript, I don't know if it can be cited.

We agree with the reviewer that PI is not a perfect live/dead marker and that there is a possibility that some of the cells not stained with PI are indeed dead. We now clarify the limitation of our PI-based assessment of cell viability with the references (refs. 39 and 40) that addressed this issue (L315-318).

It is interesting and important to study the quantitative correlation between RpoS levels and the lag time of individual cells. However, since the functional issue of the RpoS-mCherry fusion protein has been confirmed, such an analysis requires the construction of *E. coli* strains that allow reliable quantification of RpoS levels without affecting its functions. We concluded that this was beyond the scope of this study.

The bioRxiv preprint that we cited in the original manuscript has now been published in Nature Microbiology (ref. 23). We have updated the reference in the revised manuscript.

The next section starts with the measurement of dormant cells in the different conditions tested and provides an equation to predict the frequency of surviving cells after treatment. Although I'm not competent to judge the equation, the conclusion reached by the authors is that the longer in the stationary phase the better sit is for the survival of ampicillin. Some papers from the Tenson group came to this conclusion and should be cited e.g. Luidalepp et al. J. Bact 2011; Joers and Tenson, Sci Rep 2016; Joers et al. J Bact 2010. It would certainly be interesting to discuss these data in light of the present single-cell analysis.

We thank the reviewer for pointing out the important literature. The paper, Luidalepp, *et al.* J. Bact 2011, is the most relevant to the conclusion of our analysis, and we now explicitly state the consistency of our conclusion with this reference (ref. 43) at the end of this section with a brief explanation of how they estimated the number of non-growing cells in their assay (L391-394).

Dormant cells from the late stationary phase express high levels of RpoS that seem to form aggregates. Although it's plausible, it would be nice to show some evidence. The authors could try the IbpA fusion that colocalizes with aggregates – recently published by the group of Aertsen (Govers et al., PLoS Biol 2018). The assumption here is that in the early or late stationary phase, the cells with the highest RpoS levels are not the persisters and that RpoS is aggregated in those cells. What is the idea? Do aggregates somehow prevent cells to regrow after the ampicillin treatment? Are these cells dead? Are the RpoS-specific aggregates somehow toxic? Or is it a general phenomenon evidenced here by the RpoS fluorescent fusion? In other words, if the authors use another fluorescent marker reporting a protein highly expressed in the stationary phase, will they observe the same phenomenon.In any case, it would be interesting to characterize these cells further and link the data with those presented in other papers, e.g. several papers from the Michiels group that link aggregates and persistence (Dewachter et al., mBio 2021; Bollen et al., Front Mol Biosci 2021).Figure 4F is not clear, the red dots are barely visible.

We agree that it is very interesting to test the co-localization between RpoS aggregates and IbpA aggregates in the non-growing cells in late stationary phase. However, as mentioned above, we have confirmed the problem with the RpoS-mCherry fusion. Therefore, this test will only be meaningful after establishing an appropriate reporter of RpoS localization in cells, which we again concluded was beyond the scope of this study.

The last section deals with persistence to ciprofloxacin and parallels the data observed by Goormaghtigh and Van Melderen Ref 20. In the reference list, the last name of the last author should be Van Melderen, not Melderen.

Corrected the author name of the reference.

Reviewer #3 (Recommendations for the authors):I couldn't find the information on how long cells are exposed to fresh media after antibiotic treatment. It is important for the reader that this phase is long enough, so it should be mentioned in relevant places.

We thank the reviewer for pointing this out. In our experiments, we observed cells after antibiotic treatments for typically 12 hours in LB and 24 hours in M9. However, we also note that when cells begin to regrow rapidly after treatment, their progeny fill the chamber by growth and eventually push up the semipermeable membrane covering the chambers. As a result, they grow into the neighboring chambers and prevent further observation of the neighboring chambers. In this case, we had to stop observing the invaded chambers earlier than the above-mentioned durations. We now clarify the duration of observation after antibiotic treatments in the corresponding sections of Results (L128-133) and Methods (L817-819 and L824-827) and mention the limitation caused by the progeny of rapidly growing cells where we first introduce the experimental procedure with the MCMA device (L128-133).

Several semantic issues:The word dormant is used here for "which has not restarted growing during the 2 h in LB before the treatment". This type of growth arrest where cells are already exposed to fresh media but not yet regrowing (hence more likely to have resumed some metabolic activity) is almost universally referred to as "lag" and this would be a much more appropriate wording. Dormancy is a more loosely defined term that seems to imply for most people a prolonged growth arrest.

As mentioned in our response to major comment 1 and to Essential Revision 3, we now refer to these cells simply as “non-growing cells”.

The authors often refer to persister lineages – in general one rather likes to think of persister cells which each give rise to a lineage (where most cells are expected to be sensitive).

We have changed "persister lineages" to "persister cells" in the revised manuscript.

The article is mostly about persistence to ampicillin which does not represent all antibiotics, hence the title and writing would benefit from being toned down.

We thank the reviewer for this comment. We agree that our study is limited to only two antibiotics, and have added a discussion of this limitation in the Discussion (L576-581).

The rationale for fitting the killing curve in Figure 1A with a triple exponential is unclear.

We used the Levenberg-Marquardt method to fit the killing curves in this study. We found that some of the experimental killing curves deviated significantly from the double exponential curves. Therefore, we evaluated Akaike's Information Criterion (AIC) by changing the number of exponential functions (i.e., by changing *n* of the fitting function fn(t)=∑j=1naje−kjt, where aj and kj are the fitting parameters) and selected the number that gave the minimum AIC with the constraints aj>0 and kj>0*k*.

The better fit of the multi-exponential curves to the experimental killing curves was previously reported by Svenningsen, *et al.* (ref. 46). We now explain the rationale behind the fitting in the captions of Figure 1 —figure supplement 3 and Figure 6 and also in Methods (L893-902). We also discuss the implications of the multiphasic killing curves in Discussion, citing the ref. 47 (L503-515).

By comparing Figure 3B and 3C, it is remarkable that the shortest lags in Late Stationary are shorter than the shortest lags in Early Stationary which is very counter-intuitive and likely to point to an experimental issue. In addition, the mode of the elongation rate for growing cells in Figure 3E is lower than in Figure 3D and 3F which is also unexpected.

We again thank the reviewer for carefully evaluating our results. First, we note that due to the change in the main *E. coli* strain used in this manuscript from MF1 to MG1655 and the way to evaluate the pre-exposure growth state, the previous Figures 3A-F are no longer shown in the revised manuscript.

However, based on the reviewer's comment, we reexamined the single-cell data of the MF1 regrowing from the different growth phases in the microfluidic device and found errors in tracking a part of single cell lineages of MF1 and in adjusting the time frames of the measurements across the conditions. We corrected these errors and replotted the growth kinetics by summing the number of individual cells at each time point and found that those of the post-early stationary phase and post-late stationary phase cell populations are almost indistinguishable, as shown in the figure below. Therefore, the lag times under these post-early and post-late stationary phase conditions were similar in this experimental setting. Although it is interesting to analyze these lag behaviors in more detail, we judged that more careful experiments and analyses specifically designed to address this question would be necessary and are beyond the scope of the current manuscript.

**Author response image 1. sa2fig1:** Regrowing population kinetics of MF1 from different growth phases.

In contrast to most manuscripts I review, this manuscript was exceptionally well proofread and features virtually no typos. I only spotted " the same protocol as for growth curve measurements" (p. 17), as well as the caption of Supp Video 11.

We thank the reviewer for carefully reading our manuscript. We have corrected these typos.

Coming back to the first major concern expressed in the Public Review, I explain hereafter why I believe that "dormant" cells are expected to possibly be in very different states in different conditions (i.e. they do not have the same state in terms of gene expression, nor of metabolic state). I also comment on what is expected following Balaban et al. 2019; note that I assume for this that the state leading to spontaneous persistence also happens after the stationary phase and that persisters in the stationary phase can hence be decomposed into spontaneous and triggered. Importantly this has not been established and another description might well be more relevant. The 4 main conditions studied here are:Exponential (diluted):Preculture: 6 h growth in LB after ~5e5x dilution (hence 15-16 divisions) from overnight culture (5 to 10 h in stationary phase).Assay: 1.5 h growth in LB followed by 3.5 h treatment. This condition is designed specifically to get rid of non-growing cells which might have been carried over from the last stationary phase; hence only spontaneous persistence is expected.Exponential:Preculture: 2.5 h growth in LB after ~500x dilution (hence 6-7 divisions) from overnight culture (5 to 10 h in stationary phase).Assay: same (1.5 h growth in LB followed by 3.5 h treatment). Given the relatively low dilution factor from the overnight culture, one expects a mixture of spontaneous persistence and possibly some triggered persistence carried over from the last stationary phase.Early Stationary:Preculture: 8 h growth in LB after ~500x dilution (hence 4 h in stationary phase) from overnight culture (5 to 10 h in stationary phase).Assay: 2 h growth in LB followed by 3.5 h treatment.Since most cells restart growing during the 2 h in LB at the start of the assay (Figure 3E), one does not expect tolerance by lag to play a role; however since cells divide only 1 to 3 times since the last stationary phase, triggered persistence is expected to occur at a higher frequency than spontaneous persistence.Late Stationary:Preculture: 24 h growth in LB after ~500x dilution (hence 20 h in stationary phase) from overnight culture (5 to 10 h in stationary phase).Assay: same (2 h growth in LB followed by 3.5 h treatment).For a quarter of the population that has not exited the lag during the 2 h in LB at the start of the assay (Figure 3F), tolerance by lag is expected to be the main mode of survival. For the other three quarters, triggered persistence is expected to happen at a higher frequency than spontaneous persistence (note that the rate is probably different from Early Stationary).Overall non-growing cells seen in Exponential (diluted) are expected to be only due to "bad luck" (Levin, Concepción-Acevedo, and Udekwu 2014). For the Exponential condition, non-growing cells are expected at a similar frequency due to "bad luck", as well as possibly due to carry over from the previous stationary phase (very long lag); note that such carryover doesn't seem to change the frequency of growing and non-growing persisters (Figure 1 of this review).In the Early Stationary condition, most cells have restarted growing only recently hence some proteins/metabolites involved in persistence are expected at a higher level, which will lead to triggered persistence. Finally growing cells in the Late Stationary condition have spent longer in the stationary phase, which means that other proteins/metabolites might be present and/or at higher levels; this manifest in their longer lags before restating growing in LB (visible from the higher fraction of non-growing cells after 2 h).Regarding Late Stationary, I anticipate that the authors may claim that it cannot correspond to tolerance since the killing curve (yellow in Figure 2B) doesn't display a decrease in the initial death rate which is the hallmark of tolerance. I indeed don't know how to reconcile this observation with Figure 3F which shows that 26% of the population is still in the lag phase after the stationary phase; however since this high fraction stands as the main difference between Early Stationary (Figure 3E) and since non-growing bacteria are notoriously less sensitive to β-lactams such as ampicillin, one can only be tempted to hypothesize that these cells are indeed tolerant. In fact, I would like to suggest that maybe the killing curve would look completely different for Late Stationary (but not for the other 2 conditions) if prolonged far beyond the lag time of this strain out of the stationary phase.Balaban, N.Q. et al. (2019) 'Definitions and guidelines for research on antibiotic persistence', Nature Reviews. Microbiology, 17, pp. 441-448. doi:10.1038/s41579-019-0196-3.Goormaghtigh, F. and van Melderen, L. (2019) 'Single-cell imaging and characterization of *Escherichia coli* persister cells to ofloxacin in exponential cultures', Science Advances, 5(6), p. eaav9462. doi:10.1126/sciadv.aav9462.Levin, B.R., Concepción-Acevedo, J. and Udekwu, K.I. (2014) 'Persistence: a copacetic and parsimonious hypothesis for the existence of non-inherited resistance to antibiotics', Current Opinion in Microbiology, 21, pp. 18-21. doi:10.1016/j.mib.2014.06.016.Maisonneuve, E., Castro-Camargo, M. and Gerdes, K. (2013) '(p)ppGpp controls bacterial persistence by stochastic induction of toxin-antitoxin activity', Cell, 154(5), pp. 1140-1150. doi:10.1016/j.cell.2013.07.048.Patange, O. et al. (2018) '*Escherichia coli* can survive stress by noisy growth modulation', Nature Communications, 9(1), p. 5333. doi:10.1038/s41467-018-07702-z.To propose constructive steps forward, I want to reiterate that the experimental finding that bacteria surviving antibiotic treatment are, for a substantial fraction, growing before drug exposure is very important (although not entirely new). I hence believe that the authors are very well-positioned to propose a much more compelling interpretation of their experimental data. In particular, one would like to see a quantitative analysis of the different modes of survival. This probably requires measuring two additional datasets for Early Stationary and Late Stationary in order to disentangle tolerance from triggered persistence, namely the distributions of lag durations until bacteria restart growing in fresh media, as well as the killing curves (directly in Stationary Phase, not after 2h of growth in LB). By combining those, it will be possible to test whether the observed levels of survival can be interpreted as mixtures of tolerance and persistence, as well as to quantify the difference in frequency of triggered persistence for Early Stationary and Late Stationary. Such an analysis has the potential to be an important contribution to the field by shaping its conceptual framework more clearly, which will very likely require modifying it (one can for instance debate the nature of the continuum between triggered persistence and tolerance).

We are grateful to the reviewer for summarizing the results under different conditions and explaining the reasoning behind the previous comments.

As mentioned in our response to comment (1), we agree that non-growing cells observed under different conditions in our single-cell experiments could be in different states; we did not intend to assume that they were of the same type. We also note that we were aware of the recently introduced classification of persisters into "triggered" and "spontaneous" (Balaban, et al., 2019).

Although the reviewer's reasoning described above may be correct, it is practically difficult to conclusively classify the observed persisters into triggered and spontaneous ones at the singlecell level without making assumptions. As mentioned by the reviewer above, spontaneous persister formation can also occur in stationary phase. In addition, the factors contributing to the formation of persisters may be diverse. Therefore, in general, it would be difficult to conclude whether the persisters might have been generated by some triggers or spontaneously, without additional information regarding the intracellular molecular events leading to the formation of persisters at the single-cell level. Distinguishing tolerance by lag and different survival modes of non-growing cells is also challenging at the single-cell level because of the graded nature of tolerance among lagging cells and also among persister cells. We agree with the reviewer that it is important and intriguing to uncover the deeper diversity behind the simple classification of nongrowing and growing persisters, and that we are well positioned to address this important question. However, we have concluded that this would go beyond the scope of the current study.

Recommendation for the next version of the manuscript:Figure 1 would be clearer if the titles of panels F and I would be moved up to panels D and G. Supp Video 19 might be better placed before Supp Video 12; Supp Video 20 might be better placed before Supp Video 15.

We appreciate the suggestion, and have changed the previous Figure 1 as recommended (Figure 2 in the revised manuscript). Supplementary Videos 19 and 20 have been removed from the revised manuscript due to the change in the main *E. coli* strain.

Regarding cells relaxing to a bacillus morphology after taking L-form (end of p. 8), it might be relevant to mention other works where *E. coli* does this after "loosing" its shape by growing through very narrow constrictions (Männik et al. 2009. doi:10.1073/pnas.0907542106).

We thank the reviewer for suggesting the addition of an important reference. We now cite this reference and mention a possible relevance of our observation to this previous study in Discussion (L534-544).